# TLCD1 and TLCD2 regulate cellular phosphatidylethanolamine composition and promote the progression of non-alcoholic steatohepatitis

The fatty acid composition of phosphatidylethanolamine (PE) determines cellular metabolism, oxidative stress, and inflammation. However, our understanding of how cells regulate PE composition is limited. Here, we identify a genetic locus on mouse chromosome 11, containing two poorly characterized genes *Tlcd1* and *Tlcd2*, that strongly influences PE composition. We generated Tlcd1/2 double-knockout (DKO) mice and found that they have reduced levels of hepatic monounsaturated fatty acid (MUFA)-containing PE species. Mechanistically, TLCD1/2 proteins act cell intrinsically to promote the incorporation of MUFAs into PEs. Furthermore, TLCD1/2 interact with the mitochondria in an evolutionarily conserved manner and regulate mitochondrial PE composition. Lastly, we demonstrate the biological relevance of our findings in dietary models of metabolic disease, where Tlcd1/2 DKO mice display attenuated development of non-alcoholic steatohepatitis compared to controls. Overall, we identify TLCD1/2 proteins as key regulators of cellular PE composition, with our findings having broad implications in understanding and treating disease.

The fatty acid composition of phospholipids in cell membranes affects their biophysical characteristics, such as fluidity, curvature, and permeability[1]. Furthermore, membrane phospholipid composition determines the cellular response to oxidative stress[2] and modulates the production of lipid signaling mediators[3]. Phosphatidylethanolamine (PE) is the second most abundant mammalian phospholipid, and is enriched in mitochondria and the inner layer of the plasma membrane[4]. PE composition has been shown to affect cellular metabolism[5], oxidative stress[2,6] and inflammatory signaling[7–9] in both cell and animal models.

Our knowledge on how PE composition is regulated is limited to the sn-2 fatty acyl chain position. Specifically, the enzymes ACSL4 and LPCAT3 act concordantly to incorporate polyunsaturated fatty acids (PUFAs) into the sn-2 position of PEs[2,10]. PUFA-containing PEs are highly susceptible to peroxidation, which can lead to ferroptotic cell death[2,6].

Furthermore, PEs containing PUFA at the sn-2 position are precursors for the production of eicosanoids, which are bioactive signaling molecules involved in diverse processes including inflammation[11,12]. At the sn-1 position, PEs typically contain either saturated fatty acids (SFAs) or MUFAs[13]. However, the mechanisms that regulate the fatty acid type at this position, and how that influences its biological functions are presently unknown.

In this work, we utilize mouse genome-wide association data, loss-of-function mouse and human cell models, and lipidomic techniques to establish TLCD1 and TLCD2 as key regulators of cellular PE composition. We show that TLCD1/2 proteins promote the incorporation of MUFAs into the sn-1 position of PEs. We further demonstrate that human TLCD1/2 and their *Caenorhabditis elegans* homolog FLD-1 interact with mitochondria, and that TLCD1/2 regulate mouse hepatic mitochondrial PE composition. Finally, we show that Tlcd1/2 DKO mice

✉ e-mail: kasparas.petkevicius@astrazeneca.com

**Fig. 1 | *Tlcd1* and *Tlcd2* genes regulate the levels of SFA- and MUFA-containing PE species in mouse liver. a** Quantitative trait loci for all PE species measured by LC–MS/MS in the total liver of 8 different founder strains ($n = 8$ mice/strain) and 384 diversity outbred mice, then mapped onto the mouse genome. This reveals the genomic position and founder strain allele effect pattern for each PE species (data from Linke et al.[14]) **b** LOD score plots for selected hepatic PE species across the indicated region in mouse chromosome 11 (graphs adapted from lipidgenie.com). The position of *Tlcd1/2* genes is indicated with dotted lines. **c** Volcano plot of lipid species measured with high confidence (490 species in total, PC indicated in red and PE in blue) in wild-type and Tlcd1/2 DKO chow-fed, 3-month-old male mouse livers ($n = 10$ mice/group). **d** Total measured PE levels and molar percentage of MUFA and

SFA fatty acyl chains in PE species of the same mice as in **c**. Separate analysis was performed for PE acyl chain composition ($n = 8$ mice/group). **e** The levels of PE with 16:1 and 20:4 acyl chains, expressed as molar percentage of total PE species, measured in chow-fed 3-month old male ($n = 10$ mice/genotype) and female ($n = 10$ mice/ genotype), 10-month old male ($n = 9$ WT, 7 DKO) and female ($n = 10$ WT, 9 DKO), 18-month-old male (8 WT, 9 DKO), as well as 10-month old male ($n = 12$ WT, 13 DKO) and female ($n = 13$ WT, 9 DKO) fed WD for 32 weeks. In **c, d**, data are presented as mean values ± SEM. In **c**, the logarithms of multiple unpaired two-tailed student's *t*-test *p* values (not adjusted for multiple comparisons) are plotted on the *y* axis. In **d**, two-tailed student's *t*-test *p* values are indicated on graphs. In **e**, statistical differences were not assessed. Source data for **a, c–e** are provided as a Source Data file.

fed a high-fat diet (HFD) or a Western diet (WD) have attenuated development of fatty liver disease and nonalcoholic steatohepatitis (NASH), respectively.

## Results

### TLCD1/2 regulate the levels of SFA- and MUFA-containing PE species in mouse liver

To determine how PE composition is regulated at the genomic level, we specifically looked for quantitative trait loci of all measured PE species in liver in a large-scale genome-lipid association map[14] (see "Methods"). By far the strongest association with numerous hepatic SFA- and MUFA-containing PE species was observed on chromosome 11 (Fig. 1a). Our search for candidate genes that could regulate PE levels in this locus revealed two genes of poorly characterized functions, *Tlcd1* and *Tlcd2* (Fig. 1b). These two genes share a high sequence similarity and encode transmembrane proteins containing TRAM-Lag1p-CLN8 domains (TLCD), which are involved in lipid sensing, trafficking and metabolism[15]. The closest *C. elegans* homolog of *Tlcd1/2* was recently shown to regulate fatty acid incorporation into membrane phospholipids in worms, where PE is the most abundant phospholipid species[16]. Both *Tlcd1* and *Tlcd2* are highly expressed in mouse liver[16], and their

expression levels are enriched in hepatocytes (Supplementary Fig. 1a). Furthermore, similar to other genes involved in de novo lipogenesis (DNL) and lipid remodeling[17,18], mouse hepatic *Tlcd1* and *Tlcd2* expression is suppressed in the fasted state (Supplementary Fig. 1b).

To test whether the *Tlcd1/2* genes are involved in PE homeostasis, we generated Tlcd1 knockout (KO), Tlcd2 KO and Tlcd1/2 DKO mice (Supplementary Fig. 2a, b) and compared their hepatic lipidomes to the respective 3-month-old wild-type controls. Alterations in hepatic PE composition were observed in both Tlcd1 KO and Tlcd2 KO mice, with *Tlcd1* deletion resulting in major reductions in MUFA-containing PE species (Supplementary Fig. 1c). In contrast, *Tlcd2* deletion reduced the levels of only two MUFA-containing PE species (Supplementary Fig. 1d). Tlcd1/2 DKO mice exhibited a strong decrease in hepatic PE species containing palmitoleate (16:1 MUFA) and oleate (18:1 MUFA) at the sn-1 position (Fig. 1c, Supplementary Fig. 1e).

To avoid any potential functional redundancies between *Tlcd1/2* genes, and as the deletions of either gene affected PE composition without major effects on other hepatic lipid species, we focused our further investigation on Tlcd1/2 DKO animals. Fatty acyl composition analysis showed that MUFAs in hepatic PEs were reduced at the expense of increased SFAs in PEs in Tlcd1/2 DKO mice compared to

controls (Fig. 1d, Supplementary Fig. 1e). Furthermore, MUFA-containing hepatic PE species were depleted in Tlcd1/2 DKO animals regardless of sex, age, or diet (Fig. 1e).

To test whether the global *Tlcd1/2* deletion had any physiological consequences in mice, we performed a thorough phenotyping of Tlcd1/2 DKO animals. Tlcd1/2 DKO mice developed normally and exhibited an unaltered phenotype in standard behavioral and physiological tests (Supplementary Fig. 3a, b). No changes in energy balance were observed between chow-fed Tlcd1/2 DKO and control animals at either room temperature or thermoneutral housing conditions (Supplementary Fig. 3c, d). Tlcd1/2 DKO mice also had unaltered glucose tolerance on a regular chow diet (Supplementary Fig. 3e, f). Overall, altered hepatic PE composition in Tlcd1/2 DKO mice did not have any major effects on animal physiology and metabolism under standard housing conditions.

## TLCD1/2 promote MUFA incorporation into the sn-1 position of PE in a cell-intrinsic manner

To investigate how TLCD1/2 regulate PE composition, we first evaluated whether they mediate the intracellular uptake of MUFAs by gavaging mice with MUFA-radiolabeled triolein. Both control and Tlcd1/2 DKO groups accumulated radiolabel in serum and tissues at the same rate, indicating no effect on MUFA uptake (Supplementary Fig. 4a, b). We also evaluated whether the effects of *Tlcd1/2* deletion on hepatic PE composition were mediated by altered hepatic lipid metabolism. However, we observed similar changes in PE composition between Tlcd1/2 DKO and control mice in both fasted and refed states (Supplementary Fig. 4c, d). To determine whether TLCD1/2 remodel hepatic PEs by regulating the transcription of genes involved in lipid remodeling (i.e., *Lpcat*, *Lpeat* or *Lpaat* family members[19]), we compared the hepatic transcriptome of Tlcd1/2 DKO to wild-type mice. Tlcd1/2 DKO mice exhibited an unaltered hepatic transcriptome compared to controls (Supplementary Fig. 5a). This also suggests that Tlcd1/2 DKO-driven hepatic lipidome changes were well-tolerated under standard housing conditions. To further establish the post-transcriptional mechanism of action of TLCD1/2 proteins, we measured PC and PE species in red blood cells, which lack gene transcription, and thus acquire their phenotypes post-transcriptionally[20]. Indeed, red blood cells isolated from Tlcd1/2 DKO mice had similar alterations in PE composition as those observed in the liver (Fig. 2a).

By performing a time-course of primary mouse hepatocytes isolated from Tlcd1/2 DKO mice, we discovered that they maintained their altered PE composition after 2 days of culture, implying cell-intrinsic effects of TLCD1/2 proteins (Fig. 2b). Thus, we performed a pulse-labeling experiment in cultured hepatocytes to assess the rate of SFA and MUFA incorporation into cellular lipids (Fig. 2c). Tlcd1/2 DKO hepatocytes showed a strongly reduced rate of MUFA incorporation, and an increased rate of SFA incorporation into the sn-1 position of PE species compared to controls (Fig. 2d). In contrast, there was no difference in the rates of incorporation of either SFA or MUFA into the sn-1 position of PC species between genotypes, demonstrating a specific effect on PE (Fig. 2d). Both genotypes showed similar rates of SFA- or MUFA-containing acylcarnitine labeling, suggesting unaltered mitochondrial oxidation of exogenous fatty acids in Tlcd1/2 DKO cells (Supplementary Fig. 5b). The incorporation of SFA and MUFA into TG species was largely comparable between genotypes, although Tlcd1/2 DKO hepatocytes tended to have increased abundance of double-labeled 16:0-18:1-18:1 TG species compared to controls, suggesting that MUFAs could be diverted from being incorporated into PE, to being stored as TGs in Tlcd1/2 DKO cells (Supplementary Fig. 5b).

Fatty acids are incorporated into mature phospholipid species via the Lands' cycle, by lysophospholipid acyltransferases (LPLATs) acting on fatty acyl-CoA and lysophospholipid reaction intermediates[19]. We therefore assessed the rate of both canonical LPLAT reactions and of the custom sn-2-lysoPE acyltransferase reactions, designed to produce

the PE species depleted in Tlcd1/2 DKO mice, using an in vitro assay in liver microsomes[21]. There were no differences between genotypes in any of the LPLAT assays (Supplementary Fig. 5c). Altogether, these results indicate that TLCD1/2 act cell intrinsically, and upstream of LPLATs to regulate incorporation of MUFA into the sn-1 position of PE.

We also generated three TLCD1/2 DKO clones in a human hepatocellular carcinoma line to determine whether their function in regulating PE composition is conserved in human cells (Supplementary Fig. 2c–e). Indeed, compared to the unedited pool or to a single cell-derived clone of wild-type cells, all three TLCD1/2 DKO clones exhibited strongly reduced levels of MUFA-containing PE species, accompanied by an increase in SFA-containing PEs (Fig. 2e).

## TLCD1/2 interact with mitochondria and regulate hepatic mitochondrial PE composition

To further understand the mechanism of action of TLCD1/2, we generated stable human HepG2 and HeLa (as a superior cell model for microscopy than HepG2) cell lines expressing human influenza hemagglutinin amino acid 96-106 epitope (HA)−tagged TLCD1 or TLCD2 proteins (HA tags were placed on C termini, as FLD-1, the *C. elegans* homolog of TLCD1/2, is functional with a C-terminal green fluorescent protein, GFP, tag[16]). The stable expression of HA-TLCD1 and HA-TLCD2 in HepG2 cells resulted in modest increases in MUFA-containing PE species, accompanied by a decrease in SFA-containing PEs compared to controls, with HA-TLCD1 expression having a stronger effect compared to HA-TLCD2 (Supplementary Fig. 6a).

Similar to other proteins involved in lipid remodeling[22], both HA-TLCD1 and HA-TLCD2 localized to the ER/Golgi network (Supplementary Fig. 6b). We next performed immunoprecipitation (IP) and proteomics to map the interactome of TLCD1/2 proteins and determine whether it is evolutionarily conserved by comparing it to the interactome of *C. elegans* FLD-1 (Fig. 3a). Most of the proteins enriched in the GFP-FLD-1 immunoprecipitate in worms were components of the mitochondrial respiratory chain (Supplementary Data 2a). Similarly, in the HA-TLCD1 and HA-TLCD2 IPs, we found an enrichment of proteins involved in oxidative phosphorylation and mitochondrial function, including three homologs of the ATP synthase subunits that were also identified in the GFP-FLD-1 IP (Fig. 3a, b, Supplementary Data 2a–c). Furthermore, several proteins involved in phospholipid synthesis and lipid remodeling were among the top HA-TLCD1/2 interacting partners (Supplementary Fig. 6c). These included PHB and PHB2, which form a mitochondrial lipid remodeling complex[5]. Most mitochondrial phospholipid metabolism takes place at the mitochondria-associated membranes where PE is enriched[23]. Indeed, HA-TLCD1 and HA-TLCD2 localized close to the mitochondria, likely at the mitochondria-associated membranes (Fig. 3c). Importantly, TLCD1 and TLCD2 are not listed in any of the databases containing the validated mammalian mitochondrial proteome, indicating that they are not intra-mitochondrial proteins[24–26].

Next, we analyzed the lipid composition of mitochondria isolated from the livers of Tlcd1/2 DKO and control mice. As expected, isolated hepatic mitochondrial fractions were enriched in the mitochondrial protein markers VDAC and COX IV (Supplementary Fig. 6d). While the levels of ER protein markers Calnexin and Calreticulin were lower in the mitochondrial fractions compared to the total liver, they were not completely depleted from the mitochondrial fractions, indicating a residual ER presence in the isolated mitochondria (Supplementary Fig. 6d). Similar to the total liver lipidome, Tlcd1/2 DKO mitochondria had diminished levels of MUFA-containing PE species, at the expense of increased SFAs at the sn-1 position of PEs (Fig. 3d). Mitochondrial PE is synthesized from phosphatidylserine (PS) precursors[27], but no changes in several measured mitochondrial PS species were observed between genotypes (Supplementary Fig. 6e). Similarly, the abundance of PS species measured in the whole liver were comparable between the wild-type and Tlcd1/2 DKO mice (Supplementary data 1c). Overall,

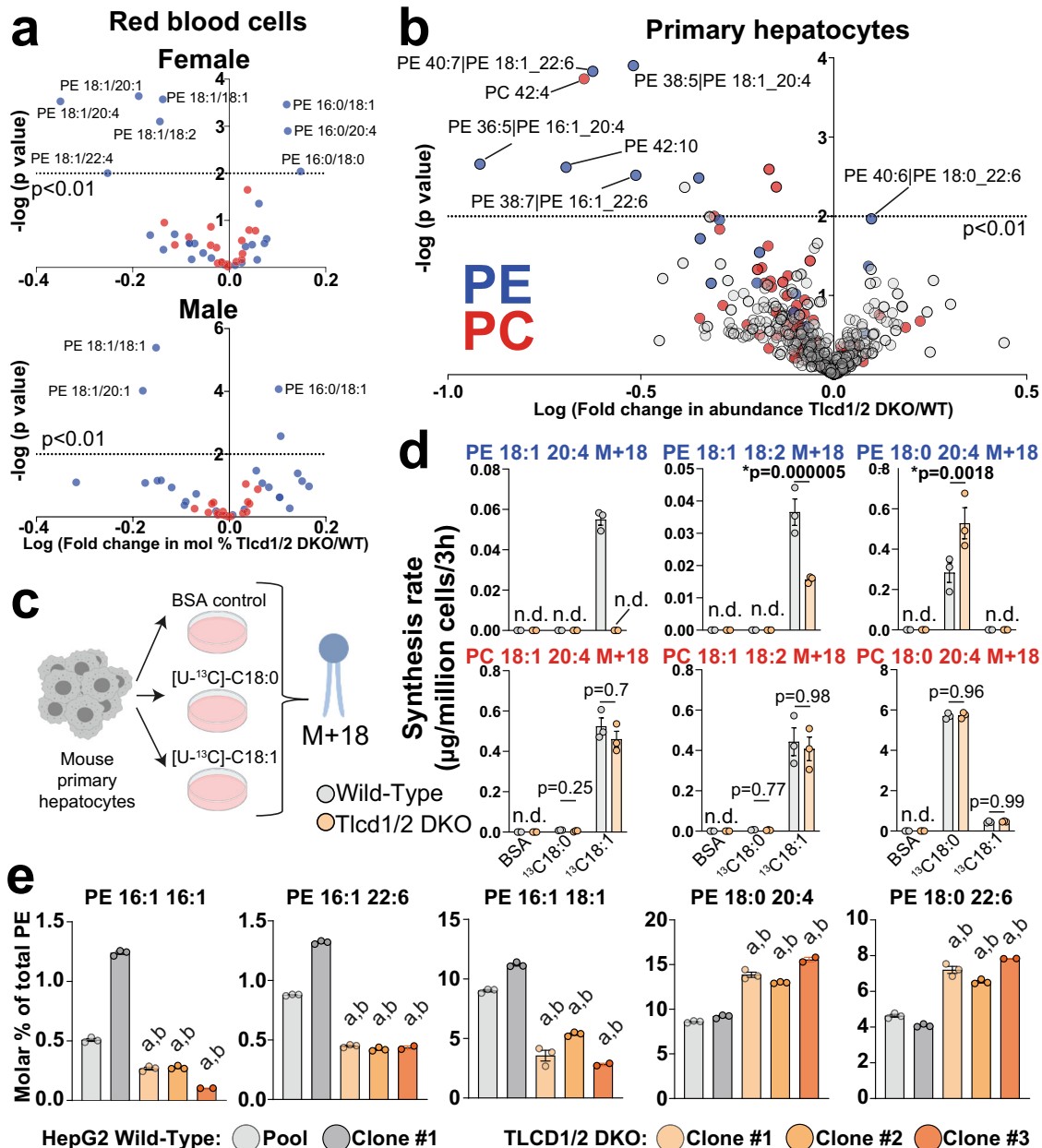

**Fig. 2 | TLCD1/2 promote MUFA incorporation into PE species in a cell-intrinsic manner. a** Volcano plots of selected PC (19 species, red) and PE (26 species, blue) species measured in red blood cells isolated from 8-month-old male (n = 12 WT, 11 Tlcd1/2 DKO) and female (6 WT, 9 Tlcd1/2 DKO) mice. **b** Volcano plot of lipid species measured with high confidence (492 species in total, PC indicated in red and PE in blue) in primary hepatocytes isolated from Wild-type and Tlcd1/2 DKO chow-fed, 3-month-old female mice (n = 3 mice/genotype) after 2 days of culture. **c** Outline of PE pulse-labeling experiment. **d** The levels of indicated PE and PC species containing stable label (mass + 18 Da) in primary hepatocytes isolated from wild-type and Tlcd1/2 DKO chow-fed, 3-month-old female mice (n = 3 mice/genotype), treated with 100 μM [U-13C]−18:0 or −18:1 for 3 h. **e** The levels of indicated PE

species (expressed as molar % of total measured PE) measured in a pool or single clone-derived HepG2 cells transfected with non-targeting gRNA, and in CRISPR-edited TLCD1/2 DKO three single clone-derived cell populations. N = 3 technical replicates per sample. In **d**, **e**, data are presented as mean values ± SEM. In **a**, **b**, the logarithms of multiple unpaired two-tailed student's t-test p values (not adjusted for multiple comparisons) are plotted on the y axis. In **d**, two-way ANOVA Sidak's multiple comparisons post-hoc test p values are indicated on graphs, and n.d. indicates undetectable levels of measured lipid. In **e**, a indicates p < 0.0001 vs wild-type pool, and **b**−p < 0.0001 vs wild-type clone 1 using one-way-ANOVA with Sidak's multiple comparisons post-hoc test. Source data for **a**, **b**, **d**, **e** are provided as a Source Data file.

our data revealed that TLCD1/2 and their *C. elegans* homolog FLD-1 interact with mitochondria, and that TLCD1/2 regulate mouse hepatic mitochondrial PE composition.

**Tlcd1/2 DKO mice show attenuated development of diet-induced liver disease**

Defective hepatic mitochondrial PE metabolism is implicated in the development and progression of nonalcoholic fatty liver disease

(NAFLD)[28]. To test whether the *Tlcd1/2* deletion-induced changes in the mitochondrial PE composition had an impact on liver disease development, we initially modeled NAFLD-associated lipotoxicity and inflammation in vitro by challenging cultured mouse primary hepatocytes with pro-inflammatory activators LPS endotoxin and the SFA palmitate (PA)[29]. Combined LPS and PA stimulation induced pro-inflammatory cytokine transcription and secretion in cultured hepatocytes without altering the expression of core hepatocyte marker

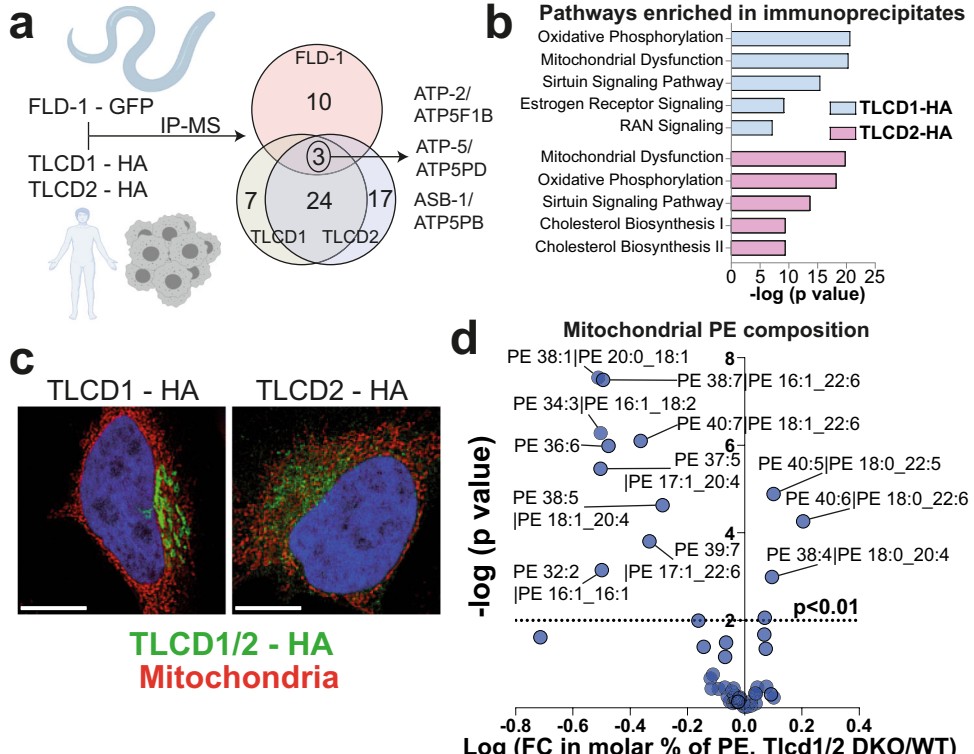

**Fig. 3 | TLCD1/2 interact with mitochondria and regulate hepatocyte mitochondrial PE composition. a** Design and summary of GFP-FLD-1 and HA-TLCD1/2 IP experiments performed in *C. elegans* GFP-FLD-1 strains (*n* = 4 biological replicates) and in HepG2 cells with stable HA-TLCD1 or HA-TLCD2 expression (*n* = 5 clones/condition). IP-MS: immunoprecipitation followed by mass spectrometry. Overlapping hits (ATP synthase subunits) are indicated. **b** Ingenuity Pathway Analysis (using genome-wide proteome reference database) of the interacting proteins enriched in either HA-TLCD1 or HA-TLCD2 HepG2 IPs compared to negative controls. The list of protein hits for each pathway are provided in the Supplementary data 2. **c** Representative (from 2 independent experiments, each containing >100 imaged individual cells) IHC images of HeLa cells with stable HA-TLCD1 and HA-TLCD2 expression, co-stained for HA tag (FITC) and mitochondria (Deep Red FM). White scale bar = 20 μm. **d** Volcano plot of PE species measured with high confidence (expressed as molar % of total PE species for each sample) in mitochondria isolated from the livers of wild-type and Tlcd1/2 DKO chow-fed, 3-month-old male mice (*n* = 7 mice/genotype). In **b**, *p* values were calculated by Ingenuity Pathway Analysis software. In **d**, the logarithms of multiple unpaired two-tailed student's *t*-test *p* values (not adjusted for multiple comparisons) are plotted on the *y* axis. Source data for **d** are provided as a Source Data file.

genes (Supplementary Fig. 7a). We did not observe any changes in DNL or hepatocyte marker gene expression or in cytokine production in LPS and PA-stimulated Tlcd1/2 DKO primary hepatocytes compared to controls (Supplementary Fig. 7a, b). Overall, *Tlcd1/2* deletion did not impact hepatocyte core gene expression in culture or modulate their inflammatory response in vitro.

To investigate the role of TLCD1/2 in the development of NAFLD in vivo, we subjected male and female Tlcd1/2 DKO mice to a HFD feeding protocol. In mice, HFD is well-known to induce systemic insulin resistance and hepatic steatosis, which does not progress to steatohepatitis and fibrosis[30,31]. No differences in body weight gain were observed between genotypes in either male or female groups during HFD feeding (Supplementary Fig. 8a). Tlcd1/2 deficiency did not affect the body composition and fasting insulin levels in HFD-fed mice (Supplementary Fig. 8b, c). At the end of the study, no changes in circulating glucose or lipid levels were observed between genotypes (Supplementary Fig. 8d). However, HFD-fed Tlcd1/2 DKO mice had reduced liver weight, lower hepatic lipid content and reduced serum levels of ALT, a marker of liver damage, compared to controls (Supplementary Fig. 8e). Liver transcriptomic analysis of the female group revealed a downregulation of the DNL-regulating molecular pathways in the Tlcd1/2 DKO females compared to female controls (Supplementary Fig. 8f). While the expression of the key DNL genes was variable in both HFD-fed male and female livers, hepatic genes mediating the biosynthesis of fatty acids were downregulated in Tlcd1/2 DKO mice compared to controls (Supplementary Fig. 8g). No changes in

genes related to fatty acid metabolism in brown and epididymal white adipose tissues were observed between genotypes (Supplementary Fig. 8h, i). Overall, Tlcd1/2 deficiency attenuated NAFLD development in HFD-fed mice.

As we observed indications of attenuated NAFLD development in the HFD-fed Tlcd1/2 DKO mice compared to controls, we hypothesized that Tlcd1/2 deletion may also be protective against NASH in mice. To test this, we fed Tlcd1/2 DKO and control mice a WD for 32 weeks to promote NASH development[30]. No changes in weight gain were observed between genotypes during the experiment (Supplementary Fig. 9a). After 30 weeks of WD feeding, Tlcd1/2 DKO animals had comparable body composition and glucose tolerance to controls (Supplementary Fig. 9b–d). However, Tlcd1/2 DKO mice exhibited reduced liver size, hepatic inflammation and fibrosis, and serum triglycerides compared to wild-type controls (Fig. 4a, c–e, Supplementary Fig. 10i, j). Similar to what we observed in HFD-fed mice, circulating ALT was reduced in the WD-fed Tlcd1/2 DKO male group compared to wild-type male mice (Fig. 4b). As expected, no differences in hepatic parameters were observed between genotypes in chow-fed controls (Supplementary Fig. 10a–e). *Tlcd1/2* deletion did not affect the levels of hepatic lipids or serum cholesterol in WD-fed mice, but it did reduce the overall incidence of hepatic macrovesicular steatosis (Supplementary Fig. 10f, h). Hepatic gene transcripts related to inflammation, macrophage and neutrophil infiltration, fibrosis and eicosanoid production were upregulated in wild-type WD-fed mice compared to wild-type chow groups and reduced in WD-fed Tlcd1/2

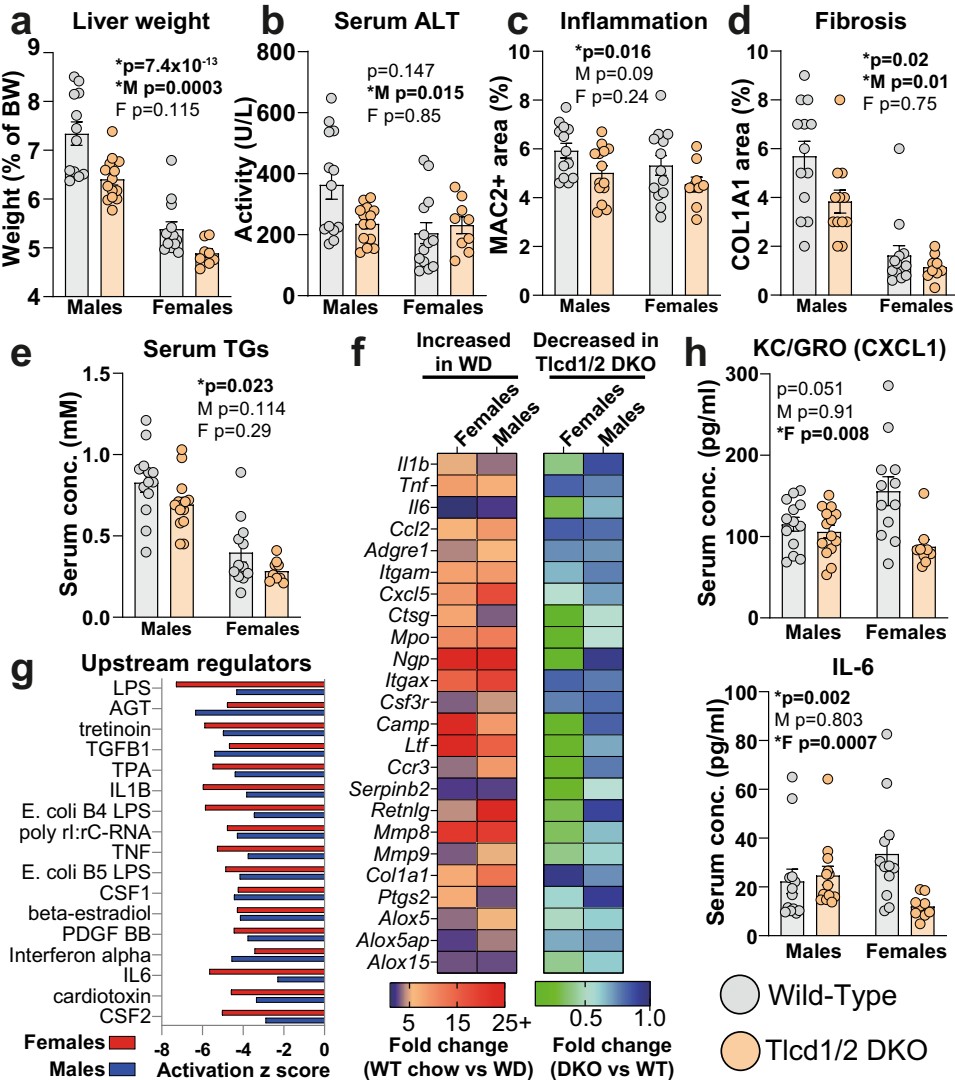

**Fig. 4 | Tlcd1/2 deficiency reduces the severity of NASH in WD-fed mice. a** Liver weights (normalized to body weights), **b** circulating ALT activity, **c** quantification of hepatic MAC2+ histological staining, **d** quantification of hepatic COL1A1+ histological staining, **e** circulating triglyceride levels in 10-month-old male and female wild-type and Tlcd1/2 DKO mice, fed WD for 32 weeks. **f** The ratios between the average expression values of indicated hepatic genes in male and female groups. Left heatmap shows the ratios of WD/chow wild-type group average expression values, and the right heatmap—the ratios of Tlcd1/2 DKO/wild-type WD group average expression values. **g** Ingenuity Pathway Analysis of predicted upstream regulators that are differentially activated in Tlcd1/2 WD-fed mouse livers compared to respective controls. **h** Serum levels of pro-inflammatory cytokines KC/GRO and IL-6 in WD-fed mice. In **a–e** and **h**, data are presented as mean values ± SEM. In all panels, $N = 9$ chow-fed wild-type males and 10 females, 7 Tlcd1/2 DKO males and 9 females; 13 WD-fed wild-type males and 13 females, 14 Tlcd1/2 DKO males and 9 females. $P$ values are indicated on each graph for genotype factor in 2-way-ANOVA, with genotype differences in male and female groups evaluated using Sidak's multiple comparisons post-hoc test. Source data for **a–f** and **h** are provided as a Source Data file.

DKO animals compared to WD-fed wild-type controls (Fig. 4f). In contrast to what we observed in HFD-fed mice, no changes in the expression of genes related to lipid metabolism were observed between genotypes (Supplementary Fig. 10k). Upstream regulator analysis demonstrated a downregulated transcriptional response to signaling by multiple pro-inflammatory cytokines and LPS endotoxin in Tlcd1/2 DKO mice compared to WD-fed controls (Fig. 4g). Furthermore, WD-fed Tlcd1/2 DKO female mice had lower levels of circulating pro-inflammatory cytokines KC/GRO and IL-6, and smaller spleens compared to the WD-fed wild-type females, indicating reduced systemic metaflammation (Fig. 4h, Supplementary Fig. 10g).

Lastly, as cell membrane PE can be a source of substrates for eicosanoid biosynthesis[32], and as hepatocyte-derived eicosanoids are implicated in NASH pathogenesis[33], we measured a panel of eicosanoids in the WD-fed mouse livers. None of the 19 detected eicosanoid species had different levels in Tlcd1/2 DKO livers compared to controls

(Supplementary Fig. 11a). However, PUFA-containing PCs and PEs, which are precursors for eicosanoid production, exhibited altered levels in WD-fed Tlcd1/2 DKO livers compared to controls (Supplementary Fig. 11b). Altogether, these results reveal that *Tlcd1/2* deletion reduces the severity of NASH in mice in a sex-specific manner, with decreased liver damage and fibrosis observed in males, while a more prominent suppression of liver inflammation was found in females.

## Discussion

Here, we discover that the fatty acyl chain type at the sn-1 position of PE is regulated by the TLCD1/2 proteins, which promote the formation of MUFA-containing PE species. Our results indicate that TLCD1/2 act downstream of cellular MUFA uptake and upstream of LPLAT enzymes incorporating MUFA-CoAs into PE species. This suggests a potential TLCD1/2 role in intracellular lipid trafficking. Interestingly, a recent study found that yeast synthesize PE containing two MUFA moieties in

the mitochondria from PS precursors[34]. Genetic yeast models in which ER-to-mitochondria lipid transport is disrupted have reduced MUFA-containing PE species at the expense of increased SFA-containing PEs[34], similar to what we observe here in TLCD1/2-deficient mice and human cells. As TLCD1/2 localize in the ER/Golgi network and interact with the mitochondria, it is plausible that they mediate the lipid transport system described in the yeast models[34], in mammalian cells. A role for TLCD1/2 in lipid transport is also supported by our data from the genetic HepG2 cell models, where the double knockout of TLCD1/2 genes strongly reduced MUFA-containing PE species, while the stable expression of HA-TLCD1 and HA-TLCD2 had a modest opposite effect, indicating that TLCD1/2 are required, but may not be rate-limiting for MUFA-containing PE biosynthesis. Additionally, the closest TLCD1/2 mammalian paralog CLN8 has been described to mediate the molecular transport of lysosomal proteins between ER and Golgi compartments[35]. Based on the publicly available mitochondrial protein databases[24–26], TLCD1/2 are not intra-mitochondrial proteins, and our results indicate that TLCD1/2 interact with only a subset of cellular mitochondria (Fig. 3c), and that this interaction may be transient. Overall, future work to further describe the interactions between TLCD1/2 and mitochondria and their potential role in mediating inter-organelle lipid transport is warranted.

We observed substantially stronger effects of *Tlcd1* deletion, compared to *Tlcd2* deletion, on hepatic PE composition. This indicates that TLCD1 is the main TLCD protein regulating PE composition, and that TLCD2 might have additional functions that are yet to be identified. It will be of interest to investigate TLCD1 and TLCD2 individually in the future to understand the differences in their mechanism of action, and their relative contributions to the liver disease phenotypes observed in the Tlcd1/2 DKO mice in this study.

Our results also implicate TLCD1/2-mediated PE remodeling in the development of NAFLD and the progression of NASH. While previous studies have associated PE metabolism to the development of liver disease[28,36–38], it is yet unclear how hepatic PE composition affects liver disease development. Our results indicate that PE species with MUFA at the sn-1 position promote the liver disease pathogenesis in the mouse models. However, our study does not reveal the biological mechanism(s) that links the PE sn-1 acyl chain profile to the hepatic function. As the PE fatty acyl chains can be hydrolyzed to provide fatty acid substrates for eicosanoid biosynthesis[32], it is plausible that altered hepatic PE composition in Tlcd1/2 DKO mice affects hepatocyte eicosanoid production, leading to reduced NASH development in WD-fed Tlcd1/2 DKO mice compared to controls. While we did not find any differences in the levels of numerous measured hepatic eicosanoid species between genotypes in the WD study, we cannot rule out the possibility that some eicosanoid species that we could not detect using our platform had different levels in Tlcd1/2 DKO mice compared to controls. As the PE species containing eicosapentaenoic acid (EPA, 20:5) were reduced in Tlcd1/2 DKO livers compared to controls on a chow diet (Supplementary Fig. 1e), it is plausible that some EPA-derived eicosanoid species that we were not able to measure, such as 18-HEPE[39], could have altered hepatic abundance in Tlcd1/2 DKO animals. Similarly, the livers of WD-fed Tlcd1/2 DKO mice had increased levels of arachidonic acid-containing PC and PE species compared to controls (Supplementary Fig. 11b). Increased hepatic arachidonic acid incorporation to phospholipids has previously been linked to reduced pro-inflammatory eicosanoid production[12], suggesting that the abundance of some hepatic eicosanoid species, that could not be detected in our analysis, may be altered in Tlcd1/2 DKO livers compared to controls. As such, a more thorough investigation on the role of TLCD1/2 in mediating hepatocyte eicosanoid production should be conducted in the future.

We observed substantial differences in Tlcd1/2 DKO mouse phenotypes between the two different dietary models of metabolic disease. Following HFD feeding, Tlcd1/2 DKO mice had reduced hepatic lipid accumulation, which was likely driven by downregulated transcription of hepatic DNL genes (Supplementary Fig. 8e–g). As HFD-fed mice have highly elevated serum insulin levels, and as insulin is well-known to stimulate DNL gene transcription[17], one plausible explanation is that *Tlcd1/2* deficiency modulates the hepatocyte insulin response during obesity. In contrast, WD feeding does not lead to systemic insulin resistance, but causes substantial liver inflammation and fibrosis[30]. The level of hepatic lipidosis was similar between Tlcd1/2 DKO and controls in the WD NASH mouse model, suggesting that TLCD1/2 can modulate NASH progression independently of hepatic DNL gene transcription and subsequent lipid accumulation. As mouse primary hepatocytes from Tlcd1/2 DKO and control mice responded similarly to a PA + LPS in vitro challenge, it suggests that a more physiological cell culture system, such as hepatocyte organoids, macrophage or stellate cell co-culture models[40], may be required to understand the pathological mechanisms of TLCD1/2.

Furthermore, the protective phenotypes in Tlcd1/2 DKO mice observed in the NASH study were different between males and females. This could be explained by sex differences in hepatic PE metabolism. The enzyme catalyzing the conversion of PE to PC, phosphatidylethanolamine N-methyltransferase (PEMT), is transcriptionally regulated by estrogen[41]. Consequently, female mice have a greater hepatic PE to PC conversion rate than male mice[42], which has been suggested to contribute to the sex-dependent differences in the development of liver disease[43]. Besides their altered PE composition, Tlcd1/2 DKO mice also show changes in the hepatic abundance of several PC species (Fig. 1c), particularly those that contain 22:6 PUFA at the sn-2 position and have been described to be produced via the PEMT pathway[44]. Therefore, while the TLCD1/2-mediated changes in PE composition are similar in males and females, the subsequent PE to PC conversion is higher in females and may explain the sex-dependent phenotypic differences observed in our NASH study.

It is also likely that TLCD1/2 have a broader role in disease development than presented here. TLCD1 was recently shown to be upregulated in hepatocellular carcinoma, where its expression levels are linked to hepatic inflammation and poor survival outcome[45]. TLCD1 has also been observed as a strongly selective gene in Cancer Dependency Map screens (depmap.org/portal/gene/TLCD1), suggesting that some cancer cells might evolve a dependency on TLCD1-mediated PE remodeling for survival, which is otherwise dispensable in normal cells, as observed here in Tlcd1/2 DKO mice. Furthermore, *Tlcd2* expression is the main determinant of eosinophil levels during allergic inflammation in murine lung[46]. Altogether, our findings open new avenues to explore the roles of TLCD1/2 in different physiological and pathological states.

## Methods

### Mice

All mouse experiments were approved by the AstraZeneca internal committee for animal studies and the Gothenburg Ethics Committee for Experimental Animals compliant with EU directives on the protection of animals used for scientific purposes. The holding facility has received full accreditation from the Association for Assessment and Accreditation of Laboratory Animal Care (AAALAC).

Mice were kept in temperature-regulated rooms (20–22 °C), 50% relative humidity and a 12:12 h light–dark cycle with ad libitum access to regular chow pellets (R3—containing 62% carbohydrate, 26% protein, and 12% caloric energy, Lactamin, Vadstena, Sweden) and tap water. All adult males were individually housed to prevent fighting, and adult females were housed in groups of 2-5 per cage.

Tlcd1 KO, Tlcd2 KO, and Tlcd1/2 DKO mice were generated using the mRNA-guided CRISPR Cas9 technology[47]. Two synthetic guide RNAs (sgRNA) were used per gene (*Tlcd1* targeting—CAGGCGACA-GAGCACGCGC and AATCTCCACCAGCATTTCAG; *Tlcd2* targeting—GCTTCCTTCACGGCGTTTCG and GCCCACCCCGAAATCTGTTC). To

make Tlcd1/2 DKO mice, all 4 sgRNAs were used simultaneously. The ribonuclear particle (RNP) was made by mixing sgRNA and spCas9 in 100 mM KCl, 20 mM Hepes buffer, and electroporated into zona-intact C57Bl/6N mouse zygotes (Janvier Laboratories) using the BioRAD Gene Pulser X-cell ™ electroporation system (BioRAD Square wave protocol; Voltage 30 V, Pulse length 3 ms, 10 pulses, pulse interval 100 ms, 1 mm Cuvette). After electroporation, the embryos were kept at 37 °C in a water-jacketed $CO_2$ incubator in M2 medium overnight. 2 cell stage embryos were implanted into pseudo-pregnant B6D2F1/Crl females. Genomic DNA from born offspring were analyzed by fragment analysis and Sanger sequencing. Regions surrounding targeted loci in Tlcd1/2 DKO mice were further assessed by Targeted Locus Amplification technology (Cergentis)[48], which revealed no off-targets or genomic translocations of targeted loci.

All generated mouse lines were backcrossed with C57Bl/6N mice (Charles River) for at least 5 generations before characterization. All experimental mice were generated by intercrossing heterozygous mice, generating male and female knockouts and littermate Wild-Type controls (due to the proximity of *Tlcd1* and *Tlcd2* genes on mouse genome, Tlcd1/2 DKO mice could be bred as a single KO model). Heterozygous offspring from each intercross were first mated to the Wild-Type C57Bl/6 N mice (Charles River), and the heterozygous offspring from that mating were used for the following intercross.

No differences between adult Tlcd1 KO, Tlcd2 KO, and Tlcd1/2 DKO mice and their respective Wild-Type controls were found with regards to general health, possible dehydration, conditioning of whiskers, eyes, blink reflexes, teeth, mucous membranes, nose discharge, paw pads, fur fitness, posture, movement (walking), feces, mobility, rearing, grooming, nesting, sniffing, digging, climbing, passivity and activity in the cage.

## In vivo studies

General phenotyping of Tlcd1/2 DKO mice was performed on adult males. Vocalization was scored after the first handling of mice by the scruff of the neck (0- silent, 1- squeak). Mouse balance and boldness was assessed using custom-made elevated balance beam, with mice placed on it for 3 min and scored 0 for falling, 1 for sitting and 2 for walking. Mouse learning in response to electric shock was assessed using shuttle box, with the power set to 0.3 mA, adaptation time 1 min, and test time up to 5 min (Shutlfx v2.10, Accuscan instruments). Mouse neuromuscular functions were assessed using Grip Strength Test (BIO-GS3, Bioseb). Heart rate was measured using ECGenie system (Mouse Specifics Inc.). Forced swim test was performed in a custom-made system, where mice were made to swim for 6 min 20 sec and their movement was tracked by camera recording.

Mouse energy balance was measured in male mice by CLAMS Comprehensive Lab Animal Monitoring System: Oxymax®-CLAMS (Columbus Instruments) at the temperature- and humidity-controlled room, set at either 21 °C or 30 °C, 50% relative humidity and a 12:12 h light-dark cycle. Mice were allowed to acclimatize for 24 h in CLAMS cages before recording the food and water intake, oxygen consumption and carbon dioxide production for 72 h. Mice that did not eat or drink in the CLAMS cages were removed from the experiment. At the end of the experiment, feces were collected, and total fecal energy content was measured by bomb calorimetry (C 6000, Ika). CLAMS data was analyzed using CalR software v1.3 (https://calrapp.org/)[49].

All mice used in this manuscript were terminated under isoflurane gas anesthesia and terminal blood samples were collected via retro-orbital bleeding. Livers were immediately excised, and the large lobe was divided into pieces, which were immediately snap-frozen in liquid nitrogen for future analyses. Approximately 50 mg of liver was used for lipidomics analyses.

For fasting and refeeding studies, mice were fasted overnight for 12 h. Next morning, mice were either terminated in the fasted state or refed with a chow diet and terminated 6 h post refeeding.

Oral glucose tolerance tests (oGTT) on mice fasted for 6 h were performed by administering a 300 mg/ml glucose solution by oral gavage, at a dose of 6.7 μl/g (2 g/kg). Basal blood samples for glucose (Accu-Check Mobile) and insulin measurements (90080, Crystal Chem, performed following manufacturer's protocol) were taken prior to gavage, and 15, 30, 45, 60 and 120 min after glucose administration.

Radiolabeled oleate uptake was performed by administering 10 μl/g olive oil spiked with 5 kBq/g [9,10-$^3$H(N)] triolein (NET431001MC, Perkin Elmer) to adult female mice by oral gavage. Serum samples were taken into glass capillaries at 1 and 2 h post gavage to measure specific activity. Terminal blood sample was taken 4 h post gavage, then tissues were excised and dissolved in SOLVABLE (Perkin Elmer) and specific activity was measured by liquid scintillation counting.

The analysis of mouse body composition was performed under isoflurane gas anesthesia using PIXImus Densitometer dual-energy x-ray absorptiometry (DEXA) instrument.

To induce obesity and insulin resistance, male and female mice were fed high-fat diet (D12492, Research Diets, 20% carbohydrate, 20% protein, 60% fat) from 16 weeks of age for 16 weeks. DEXA was performed at 30 weeks of age, fasting insulin measurement was taken at 31 weeks of age, and mice were terminated at 32 weeks of age.

To promote NASH development, male and female mice were fed Western diet (D12079B, Research Diets, 43% carbohydrate—high sucrose, 17% protein, 40% fat—high milk fat, 0.2% cholesterol) from 8 weeks of age for 32 weeks. DEXA was performed at 30 weeks of age, and oGTT at 31 weeks of age.

## Cell culture

HepG2 cells (HB-8065, ATCC) were maintained in Minimum Essential Medium (42360032, ThermoFisher) supplemented with non-essential amino acids (11140050, ThermoFisher), sodium pyruvate (11360070, ThermoFisher) and 10% fetal bovine serum (FBS, Sigma). HepG2 were passaged by detaching using Accutase (A1110501, ThermoFisher) and passing through a 1.20 × 50 mm, 18Gx2" needle thrice to minimize aggregation. HeLa (CCL-2, ATCC) cells were maintained in Dulbecco's Modified Eagle Medium (11965084, ThermoFisher) supplemented with 10% FBS and passaged using Trypsin-EDTA. The in-house stocks of HepG2 and HeLa lines were authenticated using STR fingerprint test and were 100% identity matched.

Primary mouse hepatocytes (MPH) were isolated from perfused mouse livers as described[50]. Cells were plated at a density of 1.4×10$^5$ cells/cm$^2$ on Collagen 1-coated plates in William's E medium (Gluta-MAX™ Supplement, 32551020, ThermoFisher), supplemented with 5% FBS, 17 ng/ml selenite, 56 μg/ml ascorbic acid, 3 μg/ml dexamethasone, 100 nM insulin (Actrapid) and used for experiments in 1–3 days after isolation. In the lipid labeling experiment, MPH were treated with 100 μM of either U-$^{13}$C stearic acid (CLM-6990, Cambridge Isotope Laboratories) or U-$^{13}$C oleic acid (CLM-460, Cambridge Isotope Laboratories) conjugated to 0.5% BSA (A8806, Merck) in complete MPH seeding medium for 3 h. To promote hepatocyte inflammation in vitro, MPH were treated with 1 μg/ml LPS (L2630, Merck) and 250 μM palmitate (P9767, Merck) conjugated to 0.5% BSA in complete MPH seeding medium for 24 h.

HepG2 and HeLa cell lines with a stable HA-TLCD1/2 expression were generated using our recently published Diphtheria Toxin (DT)-based selection method[51]. Briefly, HepG2 cells were electroporated with 3 plasmids at an equal ratio- (1) Cas9-encoding plasmid; (2) HBEGF Intron 3-targetting guideRNA-encoding plasmid; (3) An HDR repair template encoding HBEGF exons 4 and 5, followed by T2A cleavage site and either human TLCD1 or human TLCD2 coding sequences, followed by a peptide linker and 3xHA tag coding sequences. A manufacturer's protocol was followed with the following electroporation parameters: 1230 V, 20 width, 3 pulses (Neon Transfection System, Thermofisher). Three days later, medium was changed, and 20 ng/ml DT was supplemented to culture. Edited cells were selected for 7 days in DT-

supplemented medium, then cell lines were expanded and characterized under standard culture conditions.

HepG2 TLCD1/2 DKO lines were generated by electroporating RNPs containing recombinant spCas9 and 2 sgRNA pairs (TLCD1 targeting—TAGTGGAGATTGAGACGGCG and CCACGATGTCCACCG-TATCG; TLCD2 targeting—CCTGTCACTGTACCCTCAGA and ACCAGCACCAGAGCCCAGCG), simultaneously targeting human TLCD1 and TLCD2 genes. Non-targeting gRNA was used as a control (A35526, ThermoFisher). A manufacturer's protocol was followed with the following electroporation parameters: 1600V, 10 width, 3 pulses (Neon Transfection System, Thermofisher). Three days later, single cells were sorted into 96 well plates and clones were established in 3 weeks. Wild-type pool was cultured in parallel as a control for the experiments. Clones were validated by Sanger sequencing and qPCR.

Unless otherwise stated, all cell samples for lipidomics were collected under regular cell culture conditions. Confluent cells grown in 12-well plates were washed twice with ice-cold PBS and scraped into ice-cold BUME lipid extraction solution.

## Cell immunofluorescence

For immunofluorescence, live cells were stained with MitoTracker™ Deep Red FM (M22426, ThermoFisher) or CellLight™ ER-RFP, BacMam 2.0 (C10591, ThermoFisher) according to the manufacturer's protocols. Stained cells were washed with warm PBS and fixed in 4% methanol-free paraformaldehyde solution containing 1 µg/ml Hoechst 33342 for 10 min. Fixed cells were washed with PBS thrice and permeabilized using 0.01% digitonin PBS solution for 10 min. Permeabilized cells were blocked with 1% BSA, 3% FBS, 0.05% tween-20 PBS solution for 30 min, then stained with anti-HA-FITC (ab1208, Abcam) and anti-GM130-AF647 (ab195303, Abcam), both diluted in 1:200 in 1% BSA PBST, for 1 h. Stained cells were washed thrice in PBS and imaged on a robotic Yokogawa CV8000 spinning disc confocal microscope (Wako Automation) at 60× (NA.95, 2 × 2 binning) using ZYLA 5.5 sCMOS cameras (Andor Technology). Images were acquired with 16 bits image depth and 1024 × 1024 resolution, using a pixel dwell of ~1.02 µs. Images were analyzed using ImageJ version 1.47 software.

## Lipid extraction

Total lipid extraction was performed using automated butanol-methanol (BUME) method as described in detail in a recent methods paper[52]. Briefly, cells or liver tissue were homogenized in 0.5 ml butanol:methanol 3:1 solution spiked with either SPLASH (330707, Avanti) or equiSPLASH (3310731, Avanti) internal standards. Cold lipid extraction was performed by adding 0.5 ml 1% acetic acid and 0.5 ml heptane:ethylacetate 3:1 solution and mixing using Mixer Mill for 5 min at 25 Hz. After centrifugation, upper organic layer was transferred into glass vials and cold lipid extraction was repeated on the remaining lower phase. Second upper layer was transferred to the same glass vials. Solvents were evaporated under nitrogen flow and dried lipid extracts were stored at −80°C until analysis.

## Liver lipid content determination

Total liver lipid content in HFD-fed mouse livers were determined by pre-weighing glass vials used for lipid extraction to the 0.01 mg accuracy. The same glass vials containing dried lipid extracts were weighed again following the lipid extraction. Liver lipid content was calculated by dividing the weight of the dried total lipid extract by the weight of the liver sample used for extraction and expressed as a percentage.

## Lipidomics

Dried lipid extracts were reconstituted in 90:10 methanol:toluene and transferred to glass HPLC vials. Lipidomics analysis was performed using a Vanquish UHPLC coupled to an Orbitrap ID-X mass spectrometer (ThermoFisher). Chromatographic separation was performed on an Acquity CSH C18 column (1.7 µm, 2.1 mm×100 mm) (Waters Corporation). Mobile phases A and B were 60:40 acetonitrile:water with 10 mM ammonium formate and 0.1% formic acid and 90:10 iso-propanol:acetonitrile with 10 mM ammonium formate and 0.1% formic acid, respectively. Flow rate was maintained at 0.6 mg/mL for the entire duration of LC method. Gradient increased from 15% B to 30% B over 2 min, 30% B to 48% B over 0.5 min, 48% B to 82% B over 8.5 min, from 82% B to 99% B over 0.5 min, held at 99% B for 0.5 min, and equilibrated at initial conditions for 3 min. Column compartment was maintained at 65°C. Quantitation was performed by LC–MS analysis of each individual sample with an Orbitrap resolution of 60,000 with exception of isotopically labeled samples which were acquired with a resolution of 120,000. Lipid identification was performed by LC–MS/MS using HCD fragmentation and AcquireX DeepScan iterative data-dependent acquisition workflow (ThermoFisher) on a representative pooled sample from each study. Lipidomics data was analyzed using MS-DIAL v4[53]. Peak detection, adduct assignment, identification, and alignment were performed in MS-DIAL. Lipid annotations were performed using in silico fragmentation spectral library provided with MS-DIAL. Data from isotopically labeled samples was analyzed using a targeted method for lipids of interest using XCalibur Quan Browser (ThermoFisher). The abundance of each lipid was quantified by normalizing to the respective SPLASH internal standard amount and expressed relative to the tissue weight used for lipid extraction.

Experimental data presented in Fig. 1c, d was independently validated using Lipotype Shotgun Lipodomics service (Lipotype) as described in detail[54]. PE fatty acyl chain composition was determined from the data provided by Lipotype.

## LPLAT assays

LPLAT assays were performed as described in detail in a recent methods paper[21]. Briefly, mouse liver microsomes were prepared by homogenizing approximately 500 mg of liver in 10 ml homogenization buffer (10 mM Tis-HCl, 0.25 M sucrose, 1 mM EDTA, pH 7.0 supplemented with Roche protease inhibitor tablets) using Potter-Elvehjem tissue grinder with a PTFE pestle (1 min at 1000 rpm). Homogenate was transferred to ultracentrifuge tubes and sequentially centrifuged as follows: 10 min at $700 \times g$, 10 min at $8000 \times g$ and 10 min at $17,000 \times g$, all at 4 °C. The supernatant was then transferred to a fresh tube and the pellet was discarded. The supernatant was centrifuged at $105,000 \times g$, 4 °C for 45 min. The supernatant was then discarded and the pellet containing microsomes was resuspended in 10 ml of centrifugation buffer (20 mM Tris-HCl, 0.4 M KCl, pH 7.4). The suspension was centrifuged at 105,000 g, 4 °C for 5 min. The supernatant was discarded and the pellet containing microsomes was resuspended in 500 µl of 0.1 M Tris-HCl (pH 7.4). Protein concentration of the microsomal preparation was quantified by DC protein assay (Biorad) before aliquoting and storing microsomes at −80°C.

All LPLAT assays were set up in 0.1 M Tris-HCl (pH 7.4), 0.015% Tween-20, 20 µM LysoPL acceptor, 2 µM fatty acyl-CoA donor, and 0.1 µg microsomes. After incubation at 37 °C for 10 min, total lipids were extracted by the Bligh-Dyer method, and the assay products were subsequently measured by LC–MS/MS. Sn-1-acyl dominant LysoPL and 20:4-CoA were purchased from Avanti. Sn-2-acyl dominant lysoPE was prepared by digesting di-20:4-PE (Avanti) with $PLA_1$ as described[21]. $^{13}C$-labeled acyl-CoA used for sn-1 incorporation reactions were purchased from Sigma. For the 16:1-CoA incorporation to sn-2 20:4 lysoPE reaction, an unlabeled 16:1-CoA was used (Sigma).

## Sanger sequencing

Mouse Tlcd1/2 or human TLCD1/2 genomic loci surrounding edited region were amplified by PCR. Samples for sequencing were prepared using BigDye™ Direct Cycle Sequencing Kit (4458687, ThermoFisher) and sequenced in both forward and reverse directions on ABI 3730 sequencer (Applied Biosystems).

## Gene expression analysis

RNA from cells was extracted using RNeasy 96 kit following the manufacturer's protocol (Qiagen). RNA from adipose tissues and liver was extracted by homogenizing for 2 min in QIAzol lysis reagent, followed by column purification using RNeasy mini kit (Qiagen), according to the manufacturers' protocol. RNA was quantified using Nanodrop and cDNA synthesis was performed with 500 ng RNA input using High-Capacity cDNA Reverse Transcription Kit following the manufacturer's protocol (ThermoFisher).

cDNA was diluted 75-fold in RNAse-free water and stored at −20 °C. qRT-PCR was performed in a 10 µl reaction with 2 µl of diluted cDNA, 5 µl of 2× TaqMan Fast Advanced Master Mix/Power SYBR Green Master Mix (ThermoFisher), 0.5 µl of 20× TaqMan assays/0.5 µM forward+reverse primer mix (ThermoFisher) and 2.5 µl of RNAse-free water according to the default fast manufacturer's protocol (ThermoFisher). Reactions were run in duplicate for each sample and quantified using the ABI Prism 7900 sequence detection system (ThermoFisher). Duplicates were checked for reproducibility, and then averaged. A standard curve generated from a pool of all cDNA samples was used for quantification. The expression of genes of interest was normalized using BestKeeper method to the geometric average of 3 housekeeping genes (for mouse: *Hprt, Ppia* and *Tbp*; for human: *TBP, B2M,* and *POLR2A*), and data was expressed as arbitrary units or normalized to the average of control group.

TaqMan assays used in this manuscript were: *Hprt* (Mm03024075_m1), *Ppia* (Mm02342430_g1), *Tbp* (Mm01277042_m1), *Alb* (Mm00802090_m1), *Afp* (Mm00431715_m1), *G6pc* (Mm00839363_m1), *Hnf4a* (Mm01247712_m1), *Fasn* (Mm00662319_m1), *Scd1* (Mm00772290_m1), *Elovl6* (Mm00851223_s1), *Fads1* (Mm00507605_m1), *Fdft1* (Mm01598574_g1), *Sqle* (Mm00436772_m1), *Il6* (Mm00446190_m1), *Cxcl1* (Mm04207460_m1), *Il1b* (Mm00434228_m1), *Saa1* (Mm00656927_g1), *Ccl2* (Mm00441242_m1), *Tnf* (Mm00443258_m1), *Acly* (Mm01302282_m1), *Hmgcs1* (Mm01304569_m1), *Ucp1* (Mm01244861_m1), *Lpl* (Mm00434764_m1), *Cpt1b* (Mm00487191_g1), *Pparg* (Mm00440940_m1), *Cd36* (Mm00432403_m1), *TBP* (Hs00427620_m1), *B2M* (Hs00187842_m1), *POLR2A* (Hs00172187_m1). Primer sequences to measure gene editing at the mRNA level in HepG2 cells were: TLCD1 (FW: TGTTAGTGGAGATTGAGACGGC; RE: GTGGATGAAATACCCCGCAG); TLCD2 (FW: AAGTAACCTGTGGGCATGGG; RE: CCTGTCACTGTACCCTCAGAT).

## RNA-seq

RNA from mouse livers was extracted using QIAzol lysis reagent and RNeasy mini kit (Qiagen), following the manufacturers' protocol. The RNA-seq process was performed on BGISEQ-500 platform (The Beijing Genomics Institute, sequencing length: PE100). HISAT was used to align the clean reads to the reference genome (Mus_musculus, GCF_000001635.26_GRCm38.p6) and Bowtie2 was used to align the clean reads to the reference genes. DESeq2 was used to identify differentially expressed genes (DEGs), and fold change values were presented in heatmaps in the figures. Ingenuity Pathway Analysis (IPA, Qiagen) was used for pathway analysis and transcriptional upstream regulator prediction.

## *C. elegans* immunoprecipitations

*C. elegans* strains *Pfld-1::FLD-1::GFP* (from Ruiz et al.[16]) and *pfat-7::GFP(rtIs30)* were used to identify FLD-1 interacting proteins. *C. elegans* maintenance and experiments were performed at 20 °C using the *E. coli* strain OP50 as the food source. OP50 bacteria were maintained on LB plates kept at 4 °C and restreaked every 6-8 weeks, and single colonies were picked for overnight cultivation at 37 °C in LB medium, then used for seeding NGM plates. L1 synchronized worms were plated on five 9 cm plates per condition and grown for 72 h. Worms were collected with M9 buffer into a falcon tube, washed three times in M9 buffer and pelleted.

Worm pellets were lysed in 25 mM Tris/Cl (pH 7.5), 0.3 M NaCl, 0.1% NP-40, 1X HALT protease inhibitor cocktail (Thermofisher). Lysates were centrifuged at 20,000 × *g* for 20 min at 4 °C and incubated with GFP-Trap magnetic agarose beads (Chromotek) for 1 h at 4 °C with rotation. Beads were magnetically separated and washed thrice in 25 mM Tris/Cl (pH 7.5), 300 mM NaCl, 0.1% NP-40, followed by one wash in 25 mM triethylammonium biocarbonate buffer (TEAB). Proteins were eluted with 0.2 M glycine (pH 2.5) followed by magnetic separation. The eluate was neutralized with 1 M TEAB. *pfat-7::GFP* was used as a negative control for GFP-interacting partners. Experiment was performed 4 separate times to generate 4 biological replicates.

## HepG2 immunoprecipitations

To identify HA-TLCD1/2 interacting proteins, $10^7$ control, HA-TLCD1 and HA-TLCD2 stable-expressing HepG2 cells were lysed in Pierce IP lysis buffer (ThermoFisher) supplemented with cOmplete protease inhibitors (Roche). 5 separate single clone-derived HepG2 lines were used per each condition as biological replicates. Lysates were incubated on ice for 1 h, then cleared by centrifugation at 16,000 × *g* at 4 °C for 10 min. Protein concentration was determined using Pierce BSA Protein Assay kit (Thermofisher). One milligram of each lysate was incubated with 25 µl Pierce Anti-HA Magnetic Beads (Thermofisher) for 3 h at 4°C with rotation. Samples were washed thrice in Pierce IP lysis buffer using a magnet, and proteins were eluted using Elution Buffer (88838, Thermofisher). The eluate was neutralized with 1 M TEAB.

## Proteomics

The eluates from immunoprecipitation were reduced with DL-dithiothreitol (DTT, 100 mM) at 56 °C for 30 min and then processed according to the filter-aided sample preparation (FASP) method modified from Wisniewski et al.[55]. In short, reduced samples were transferred onto 30 kDa MWCO Pall Nanosep centrifugation filters (Sigma-Aldrich), washed repeatedly with 50 mM triethylammonium bicarbonate (TEAB) and once with digestion buffer (0.5 % sodium deoxycholate (SDC), 50 mM TEAB). The reduced cysteine side chains were alkylated with 10 mM methyl methanethiosulfonate (MMTS) in digestion buffer for 20 min at room temperature and the samples were then repeatedly washed with digestion buffer. Samples were digested with trypsin (0.3 µg, Pierce MS grade Trypsin, Thermo Fisher Scientific) at 37 °C overnight, and an additional portion of trypsin (0.3 µg) was added and incubated for another 3 h. The peptides were collected by centrifugation and isobaric labeling was performed using Tandem Mass Tag reagents (TMTpro 16plex, Thermo Fischer Scientific) according to the manufacturer's instructions. The labeled samples were combined into one pooled sample, acetonitrile was evaporated using vacuum centrifugation and SDC was removed by acidification with 10% trifluoroacetic acid (TFA) and subsequent centrifugation. The sample was purified using a High Protein and Peptide Recovery Detergent Removal Spin Column and a subsequent Pierce peptide desalting spin column (both Thermo Fisher Scientific) according to the manufacturer's instructions. The peptide samples were separated into 11 fractions using the Pierce High pH Reversed-Phase Peptide Fractionation Kit (Thermo Fisher Scientific), dried on Speedvac and reconstituted in 3% acetonitrile, 0.2% formic acid for LC−MS/MS analysis.

Samples were analyzed on an Orbitrap Fusion Lumos™ Tribrid™ spectrometer interfaced with an Easy-nLC1200 nanoflow liquid chromatography system (both Thermo Fisher Scientific). Peptides were trapped on an Acclaim Pepmap 100 C18 trap column (100 µm × 2 cm, particle size 5 µm, Thermo Fischer Scientific) and separated on an in-house packed analytical column (75 µm × 35 cm, particle size 3 µm, Reprosil-Pur C18, Dr. Maisch) from 5% to 12% B over 5 min, 12% to 35% B over 72 min followed by an increase to 100% B for 3 min, and 100% B for 10 min at a flow of 300 nl/min. Solvent A was 0.2% formic acid and solvent B was 80% acetonitrile, 0.2% formic acid. MS scans were performed at a resolution of 120,000 with an m/z range of 375-1500. MS/

MS analysis was conducted in a data-dependent, with top speed cycle of 3 s for the most intense doubly or multiply charged precursor ions. Precursor ions were isolated in the quadrupole with a 0.7 m/z isolation window, dynamic exclusion set to 10 ppm, and duration of 45 s. Isolated precursor ions were subjected to collision-induced dissociation (CID) at a collision energy of 30, with a maximum injection time of 60 ms. Produced MS2 fragment ions were detected in the ion trap followed by multinotch (simultaneous) isolation of the top 10 most abundant fragment ions for further fragmentation (MS3) by higher-energy collision dissociation (HCD) at 55% and detection in the Orbitrap with a resolution of 50,000 and m/z range of 100–500.

Data analysis was performed using Proteome Discoverer version 2.4 (Thermo Fisher Scientific). The raw files were matched against the Swissprot human database (January 2021) using Mascot 2.5.1 (Matrix Science, London, United Kingdom) as search engine with a peptide tolerance of 5 ppm and fragment ion tolerance of 0.6 Da. Tryptic peptides were only accepted with zero missed cleavage, mono-oxidation on methionine was set as a variable modification, methylthiolation on cysteine and TMTpro reagent modification on lysine and peptide N-terminus were set as fixed modifications. Percolator was used for PSM validation with the strict FDR threshold of 1% for peptide identification. Reporter ion intensities were quantified in MS3 spectra at a mass tolerance of 0.003 Da, using the S/N values as abundances. Only unique peptides were used for quantification and no normalization was performed within the Proteome Discoverer 2.4 workflow. Proteins passing a strict FDR threshold of 1% were exported for further analyses of changing protein levels.

### Mitochondrial isolation

Mouse livers (200 mg/sample) were homogenized using Potter-Elvehjem tissue grinder with a PTFE pestle (1 min at 1000 rpm) in BSA/Reagent A solution (89801, ThermoFisher) on ice. Mitochondria were then isolated by centrifugation according to the manufacturer's protocol for soft tissue (89801, ThermoFisher). Isolated mitochondria were snap-frozen before performing lipid extraction or resuspended in 200 μl ice-cold T-PER™ Tissue Protein Extraction Reagent (Thermo-Fisher) for Western blotting.

### Western blotting

Liver proteins were extracted by homogenizing a piece of liver (20 mg) in 500 μl ice-cold T-PER™ Tissue Protein Extraction Reagent in 2 ml centrifuge tube containing a metal ball, at 25 Hz for 1 min. Lysates were incubated on ice for 10 min, before clearing by centrifugation at $16,000 \times g$ at 4 °C for 15 min. Protein concentration in cleared liver lysates and isolated mitochondria were determined using Pierce™ BCA Protein Assay Kit, according to manufacturer's protocol (ThermoFisher).

20 μg of total liver and mitochondrial proteins were separated by SDS polyacrylamide gel electrophoresis, according to the manufacturer's protocol (NuPAGE® Bis-Tris Electrophoresis System, ThermoFisher). Proteins were transferred onto nitrocellulose membranes using Bio-Rad tank transfer system (30 V, 16 h, 4 °C). Transferred proteins on membranes were stained with Ponceau S Staining Solution (ThermoFisher) for 10 min. Membranes were then washed with TBST, blocked in 5% BSA TBST for 1 h and probed with the primary anti-VDAC (PA1-954A, ThermoFisher), anti-COX IV (ab33985, Abcam), anti-Calreticulin (ab92516, Abcam), anti-Calnexin (PA5-34754, ThermoFisher) and anti-GAPDH (MA5-15738, ThermoFisher), all diluted 1:1000 in 3% BSA TBST, for 2 h at room temperature. Bound primary antibodies were detected using peroxidase-coupled secondary anti-rabbit and anti-mouse antibodies (P0448 and P0447, Dako, Agilent), diluted 1:2500 in 3% BSA TBST, for 1 h at room temperature, followed by enhanced chemiluminescence (WBLUF0500, Millipore). Blots were exposed digitally using the ChemiDoc MP System (Bio-Rad).

### Cytokine analysis

Cytokine levels in cell culture medium (diluted fourfold) were determined using MSD V-PLEX Mouse Cytokine 19-Plex Kit (K15255D, MSD), and in mouse serum samples (diluted twofold) using MSD V-PLEX Pro-inflammatory Panel 1 Mouse Kit (K15048D, MSD), according to manufacturer's protocols.

### Serum biochemistry

Serum biochemicals were analyzing using ABX Pentra 400 instrument (Horiba Medical) according to manufacturers' protocols. Triglyceride levels were determined using Triglyceride/glycerol blanked reagent (11877771, Roche) and Seronorm™ Lipid (100205, Sero AS) was used as control. Cholesterol levels were determined using ABX Pentra Cholesterol CP reagent (A11A01634) and P-Control (A11A01654, Horiba Medical) was used as control. ALT activity was determined using a calorimetric assay (A11A01627, Horiba Medical) and P-Control was used as a control. MultiCal (A11A01652, Horiba Medical) was used as a calibrator for all assays.

### Histopathology and image analysis

After formalin fixation, dehydration, and paraffin embedding, 4 μm sections were stained for Hematoxylin Eosin (HE), PicroSirus red (PSR), and immunohistochemistry (IHC). IHC was performed on an automated Ventana Ultra system (Ventana Medical Systems, Roche Group, USA). The rat monoclonal (M3/38) antibody against MAC2 (Galectin-3, CL8942AP, Cedarlaine, 1:5000 dilution) or rabbit polyclonal antibody against COL1A1 (LS-C343921, LSBio, 1:1000 dilution) were used to detect macrophages or collagen deposits in liver sections, respectively. Antibody binding was visualized using DAB, hematoxylin was added as a nuclear counterstain.

Image analysis was performed on digital images of the HE, MAC2, and COL1A1 stained slides, using Visiopharm Integrator System software (v2019.2, Visiopharm). Lipidosis was calculated based on the unstained area (fat) on the HE stained slides and corrected to the total section area. For MAC2 and COL1A1 quantification, DAB positive area was quantified and normalized to the total section area. The areas were detected by threshold and machine learning.

Fibrosis score assessment was performed on PSR stained slides at 200x magnification, according to Kleiner et al.[56]. The definitions of fibrosis stages are: 0= no fibrosis, 0.5-1= perisinusoidal or periportal, 2= perisinusoidal and portal/periportal, 3= bridging fibrosis, 4= cirrhosis.

### Eicosanoid analysis

Eicosanoid levels in frozen mouse livers were measured by UPLC-ESI-MS/MS as described in detail[57]. Briefly, 20 mg of the frozen liver samples were scraped into tubes on dry ice. Extraction was performed in methanol containing antioxidants and internal standards as described in detail[57]. Out of 39 measured oxylipins, only those having levels in liver samples that were over 10-fold higher than in the blank control were presented in the figure.

### Identification of the genetic regulators of PE species

Genome-lipid associations in mouse liver were identified by Linke and colleagues and their methodology is described in detail in the original publication[14]. Briefly, 8 different mouse founder strains were randomly outbred to generate 384 diversity outbred offspring. Hepatic lipid species were then quantified using LC–MS/MS lipidomics in both the founder strains (64 mice in total, 8 per each founder strain) and the diversity outbred offspring. Male and female mice were used, and all mice were terminated at 21 weeks of age, following Western diet (Envigo Teklad, TD.08811) feeding for 16 weeks. Mice were then genotyped using genotype arrays, and the genotypes were used to reconstruct the eight-founder haplotype mosaic for each diversity outbred mouse. Quantitative trait loci (QTL) mapping was performed by analyzing the genome and testing for association between the eight-

state haplotype probabilities and the measured lipid levels. Likelihood ratio statistic (LOD) was computed to evaluate the significance of the genetic effect at each pseudo-marker locus. LOD scores for PE species of interest were obtained from www.lipidgenie.com.

## Statistical analysis and reproducibility

All data is represented as mean, with error bars showing standard error of the mean and the number of replicates stated in figure legends. Some data is represented as a fold change, and it is stated in legend to what value the data represented was normalized to generate the fold change. Statistical tests used are also stated in legend. A student's t-test was used to compare two groups; multiple t-tests (one per row) were used to compare lipidomics data between two groups. Where more than one factor influenced the variable being measured, two-way ANOVA was used to test for a significant effect of each factor as well as an interaction between factors, with Sidak's multiple comparisons post-hoc test used to evaluate the differences between genotype factor in each group.

All statistical tests were performed, and graphs were generated using GraphPad Prism 9 software. DNA plasmids were designed, sequencing data was analyzed, and CRISPR-targeting visualizations were made using Geneious Prime v2021.1.1 software. Graphs and figures were edited using Adobe Illustrator 2020 software. Cartoon images of organisms, cells and culture plates were obtained from BioRender.

All presented animal experiments were performed once. The key findings of our study—namely the changes in hepatic PE composition in Tlcd1/2 DKO animals—were reproducibly measured by two independent lipidomics facilities in several separate mouse cohorts over three breeding generations. All cell culture and in vitro experiments were independently replicated at least twice, with all replication attempts showing similar results to what is presented in the figures. Stable isotope fatty acid labeling primary hepatocyte experiment, and *C. elegans* and HepG2 immunoprecipitation-mass spectrometry experiments were performed once.

## Reporting summary

Further information on research design is available in the Nature Research Reporting Summary linked to this article.

## Data availability

RNA-sequencing data have been deposited to Sequence Read Archive (SRA) database and are accessible through the SRA accession number PRJNA818121. Full lipidomics and proteomics datasets generated for this study are provided as Supplementary Data 1 and 2. Source data are provided with this paper. All other relevant data are available from the corresponding author upon reasonable request. Source data are provided with this paper.

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

## Acknowledgements

*C. elegans* strain *pfat-7::GFP(rtIs30)* was a kind gift from Amy Walker. We thank the senior staff of AstraZeneca early CVRM Metabolism department—Jacqueline Naylor, Bader Zarrouki, Daniel Lindén, and David Baker for their advice, comments, and resource management. We also thank Christopher Rhodes and John Liddle from AstraZeneca postdoc committee for their advice. Thanks to AstraZeneca Animal Sciences & Technologies staff for their in vivo support. Thanks to Kerstin Magnell, Anna-Lena Loyd, Tania Baccega, Therese Admyre and Sara Torstensson for their help in generating and phenotyping mouse models. Proteomics work was carried out by Gothenburg University Proteomics Core facility, targeted locus amplification was performed by Cergentis, RNA sequencing was performed by Beijing Genomics Institute, DNA constructs were generated by GenScript, eicosanoid measurements were performed by Swedish Metabolomics Centre, histology slides were prepared by Histocenter (Sweden) and a part of lipidomics work was done by Lipotype—we thank all the aforementioned institutions for their excellent services. We also thank Life Science Editors for Editorial assistance. This project has received funding from Vetenskapsrådet (2020-03300), Cancerfonden (19 0029 Pj), Stiftelsen för Strategisk Forskning (ID16-0049), Leading Advanced Projects for medical innovation (AMED-LEAP) Grant Number 21gm0010004h990 from the Japan Agency for Medical Research and Development (AMED) and KAKENHI Grant Numbers 20K21379 and 20K15984 from the Japan Society for the Promotion of Science (JSPS). K.P. was a postdoc fellow of the AstraZeneca R&D postdoc program during this study.

## Author contributions

K.P., M.P., M.B., M.M., and X.-R.P. conceptualized the study. K.P., H.P., A.A., D.K., M.U., L.A., L.A., J.F., S.W., I.A., R.L., D.K. and H.G.-K. performed experiments and analyzed the data. K.P., M.G., E.A., L.L., R.N., and S.H. performed and analyzed lipidomics. H.K., N.K., and J.A. performed and analyzed LPLAT assays. A.-C.A. and G.P. performed and analyzed histopathology. K.M.-B. and M.B. generated mouse models. G.S. contributed technology and provided advice. M.P., M.B., M.M., and X.-R.P. supervised the study. K.P. prepared the manuscript, and other authors contributed to the review and editing.

## Competing interests

K.P., H.P., M.S.G., A.A., A.-C.A., K.M.-B., E.L.A., M.U., L.A., L.A., J.F., S.W., I.A., R.L., D.K., H.G.-K., L.L., R.N., G.P., S.H., G.S., M.B., M.M., and X.-R.P. are presently employed by AstraZeneca and may be AstraZeneca shareholders. The remaining authors declare no competing interests.

## Additional information

Kasparas Petkevicius [1,2] ✉, Henrik Palmgren[1,3], Matthew S. Glover[4], Andrea Ahnmark[1], Anne-Christine Andréasson[5], Katja Madeyski-Bengtson[2], Hiroki Kawana[6,7], Erik L. Allman[4], Delaney Kaper [3], Martin Uhrbom[1], Liselotte Andersson[8], Leif Aasehaug[5], Johan Forsström[2], Simonetta Wallin [1], Ingela Ahlstedt[1], Renata Leke[1], Daniel Karlsson[1], Hernán González-King [5], Lars Löfgren[9], Ralf Nilsson[9], Giovanni Pellegrini[8], Nozomu Kono [6], Junken Aoki[6,7], Sonja Hess[4], Grzegorz Sienski [2], Marc Pilon [3,10], Mohammad Bohlooly-Y[2,10], Marcello Maresca [2,10] & Xiao-Rong Peng[1,10]

[1]Bioscience Metabolism, Research and Early Development Cardiovascular, Renal and Metabolism, BioPharmaceuticals R&D, AstraZeneca, Gothenburg, Sweden. [2]Discovery Sciences, BioPharmaceuticals R&D, AstraZeneca, Gothenburg, Sweden. [3]Department of Chemistry and Molecular Biology, University of Gothenburg, Gothenburg, Sweden. [4]Dynamic Omics, Centre for Genomics Research, Discovery Sciences, BioPharmaceuticals R&D, AstraZeneca, Gaithersburg, MD, USA. [5]Bioscience Cardiovascular, Research and Early Development Cardiovascular, Renal and Metabolism, BioPharmaceuticals R&D, AstraZeneca, Gothenburg, Sweden. [6]Department of Health Chemistry, Graduate School of Pharmaceutical Sciences, The University of Tokyo, Tokyo, Japan. [7]Advanced Research & Development Programs for Medical Innovation (AMED-LEAP), Tokyo, Japan. [8]Clinical Pharmacology and Safety Sciences, BioPharmaceuticals R&D, AstraZeneca, Gothenburg, Sweden. [9]Translational Science and Experimental Medicine, Research and Early Development Cardiovascular, Renal and Metabolism, BioPharmaceuticals R&D, AstraZeneca, Gothenburg, Sweden. [10]These authors jointly supervised this work: Marc Pilon, Mohammad Bohlooly-Y, Marcello Maresca, Xiao-Rong Peng. ✉e-mail: kasparas.petkevicius@astrazeneca.com

