## [Peer Review File · Nature Communications]

Reviewers' Comments:

Reviewer #1:

Remarks to the Author:

Petkevicius and colleagues present a study identifying TLCD1/2 as novel regulators of cellular PE composition, i.e. PE remodeling. The authors generated TLCD1/2 KO mice (and cells), and apply various sophisticated omics techniques including lipidomics. Of note, the lipidomics method matches the standards to get reproducible data and most importantly, it is quantitative. The paper is well written, although I think it would benefit from a separation of results and discussion. It would make sense to discuss some specific points in more depth, including the relevance/impact of TLCD2 for PE composition/remodeling compared to TLCD1. I guess, the paper was transferred from another journal, where it was submitted initially as in the letter format. Overall, the methods and analyses applied are sound, the findings are novel and their functional relevance is demonstrated. I support publishing the paper after addressing the following points.

Major points that should be clarified:

1. The basis of the study is the "large scale genome-lipid association map". The authors should explain at least some methodical basics and details of the analyses for non-experts in genome associations, in the methods section and the Legend of figure 1. Not every reader wants to look deeply into Citation [12], to understand/read the associated Figures 1A and B/method.
2. "Tlcd1 deletion resulting in major reductions in MUFA-containing PE species, while Tlcd2 deletion had a similar but lesser effect". Please put this statement into the right perspective. Tlcd2 had almost no effect on PE, except for 1 species (Ext. Figure 1D).
3. I do not understand the sense of Ext. Fig. 4A-B and the statement "To determine whether TLCD1/2 remodel hepatic PEs by regulating gene transcription". Do you mean be regulating e.g. transcription of LPCATs? Please clarify.
4. I would prefer a simple barplot showing the major PE species (e.g. >1 or 5% of total) instead of the heatmap for Fig.2E. It would more clearly provide the relevant information.
5. Are there also lipidomic (PE!) data available for the HepG2 lines (over)expressing TLCD1/2? Please include.
6. Please show the expression (e.g. western blots) of some ER and mitochondrial markers, so that it is possible to evaluate the enrichment/contamination of your mitochondrial fraction. Or, show some lipidomic data supporting a high mitochondrial enrichment (in addition to PE =>cardiolipin).
7. "Phenotyping of TLCD1/2 mice": I would shift this part to the front of the paper, just when you introduce your mouse models.
8. Are there already some other factors known influencing the severity of NASH in a sex-specific manner? Please discuss.

Reviewer #2:

Remarks to the Author:

The manuscript by Petkevicius et al. entitled "TLCD1 and TLCD2 regulate cellular phosphatidylethanolamine composition and promote the progression of non-alcoholic steatohepatitis" presents results supporting a role for Tlcd genes in the pathogenesis of NASH. The authors identified two genes (Tlcd1 and Tlcd2) in chromosome 11 that are associated with the PE composition in liver tissue. Double knockouts (DKO) mice for these genes appeared to show attenuated development of NASH. The study is comprehensive and addresses an important medical subject, but authors need to provide more convincing data on the following two aspects:

1. Results shown in Figure 3E about eicosanoid production are not strong enough as rationale for investigation in NASH models, as authors wrote in page 6, lines 7-8. First, eicosanoid production by hepatocytes appears to be in nanograms, which is quite high production. Second, eicosanoids included in the analysis are not biologically active, either are inactive metabolites and not the typical eicosanoids that one could expect to be produced by hepatocytes. I am wondering if authors selected the right platform to measure eicosanoids. Eicosanoids such as PGF2alpha are highly produced by hepatocytes, same for LTB4 metabolites and other lipid mediators derived from EPA and DHA including monohydroxy fatty acids as 18-HEPE and 17-HDHA. Third, changes in eicosanoid production are mild and lack statistical significance. Authors need to perform a more

agnostic lipidomic approach including also anti-inflammatory eicosanoids such as LXA4.

2. NASH model. Wondering why authors used the western diet model instead of the high fat diet model. Several comments. First, changes are heterogeneous, sometimes differences are observed in males other in females. Second, the model used did not induced insulin resistance. Third, fibrosis needs to be confirmed by Sirius red staining. Fourth, histology needs to be evaluated by a registered pathologist. Other points to consider: ALT levels are unusually low. Also, eicosanoid generating enzymes (at least COX-1, COX.-2, 5LOX, FLAP,..) and relevant eicosanoids need to be determined in liver tissues.

Taken these two aspects together, authors should remove lines 4-10 in page 7 or provide stronger data supporting the claims.

Apart from these two critical issues, other points to consider:

1. DKO mice have decreased MUFA. They also have decreased EPA levels. This needs to be further explored, at least the composition of 18_HEPE and discussed.
2. Radiolabeled MUFAs accumulate in BAT, a critical fat depot in NASH. BAT needs to be explored in the NASH model.
3. HepG2 clones (Fig2E) are unnecessary and do not add much to the final message. Would be more relevant to check human tissues or other human material from patients.
4. What about other phospholipids apart from PE in mitochondria? BMP, PS and cardiolipin.

Reviewer #3:

Remarks to the Author:

The paper " TLCD1 and TLCD2 regulate cellular phosphatidylethanolamine composition and promote the progression of non-alcoholic steatohepatitis" contributes with a wealth of novel data regarding the biochemical and functional effects of TLCD1 and TLCD2. The functional effects of these two proteins have not previously been well characterized and the paper therefore makes a strong contribution by extensive and valid experiments using e.g. knockouts of the two proteins. Experimental data from models of mice, cultivated human cells and C. elegans have been generated using a broad range of appropriate methods. The authors clearly describe the TLCD1/2 knockout models cause reduced levels of monosaturated fatty acid containing PE species, and broadens the characterizations into e.g. PE-influenced inflammation.

1. TLCD1/2 is in the manuscript described to have links to mitochondria. Please extend the description and discussion of intracellular localization of TLCD1 and 2 with regard to mitochondria. It seems like TLCD1 and 2 are missing in the extensive (>1000) published compendium of Mitochondrial proteins (Mitocarta, Mootha lab), although they in the present paper co-precipitate with some proteins of inner mitochondrial membranes (ATP5-complex subunits) as well as other mitochondrial proteins PHB and PHB2. It might be that co-precipitation is mediated by PE of the mitochondrial membrane (?), which would indicate only indirect physical interaction, or by transient interaction during protein import into mitochondria. Please extend discussion of these topics and refer to compendium(s) of mitochondrial proteins.

2. Figure 3B describes Ingenuity Pathway analysis (IPA) data, in a rather superficial way. Assure that more detailed data of each functional sub-group are accessible to the reader, i.e. number of proteins and their identities, and possibly the size of background dataset (genomewide or practical cellular proteome?).

Minor:

Page 5 line 2. A description of the abbreviation "HA" is lacking in the manuscript and could be added in the context of sentence "cell lines expressing HA-tagged TLCD1 or 3 TLCD2 proteins....".

Likewise could also the abbreviation "GFP" be described.

“TLCD1 and TLCD2 regulate cellular phosphatidylethanolamine composition and promote the progression of non-alcoholic steatohepatitis” – Response to reviewers

General response to all reviewers

We thank all the reviewers for helping us to improve our manuscript. We found their comments to be very insightful, and have performed major revisions in order to address them.

We made the following major changes to the manuscript:

- 1) **Manuscript format and discussion.** As Reviewer #1 noted, the original version was written for a different Nature family journal as a letter, and later transferred to Nature Communications. We have now modified the manuscript to meet the formatting guidelines of Nature Communications. Specifically, we shortened the abstract and added separate Introduction and Discussion sections. Furthermore, as we are no longer constrained by the word limit of a letter format, we have substantially expanded the Discussion section to cover the points concerning **TLCD interaction with mitochondria, the relative contribution of TLCD1 and 2 proteins, eicosanoid production, and diet/sex differences in liver disease models**. Also, to maintain the emphasis on the novelty of our findings, we chose to keep the more concise 4 figure format and a short introduction, and have included all the new results as additional supplementary figures.
- 2) **NASH mouse model.** Reviewer #2 rightly pointed out that Western diet is not a good model for insulin resistance. Indeed, it is challenging to model all the pathological characteristics of a typical obese and insulin-resistant NASH patient using a single mouse model – obesogenic diets that promote systemic insulin resistance usually only lead to hepatosteatosis that does not progress to hepatitis and fibrosis, while diets that promote NASH development usually do not cause insulin resistance (PMID: 30372785).
In this project, we originally conducted a high-fat diet (HFD) experiment to study the role of TLCD1/2 in metabolic disease. However, due to technical and Covid-19-related challenges, some key *in vivo* metabolic tests could not be performed on the HFD-fed mouse cohort, and samples for histology were not collected. As such, even though we had observed an improved liver disease phenotype in the HFD study (based on the liver weight, circulating ALT levels and biochemical liver lipid content measurements), we chose to exclude it from the previous version of the manuscript, and only presented the data from the follow-up NASH study, which was designed on the basis of the main findings from the HFD study. However, after carefully considering the concerns of Reviewer #2, and decided to include the data from the HFD study as a **Supplementary Figure 8**, to preface the Western Diet study. Furthermore, to gain a better understanding of the liver phenotype in the HFD model, we **performed a transcriptomic analysis of livers isolated from the HFD-fed female group**. Overall, we find that while **both dietary challenges resulted in improved liver disease phenotypes in the Tlcd1/2 DKO mice**, the protective mechanisms are somewhat diet-dependent (i.e. HFD resulted in reduced hepatic lipid accumulation in Tlcd1/2 DKO mice compared to controls while Western Diet did not). We discuss the potential reasons for these differences in the Discussion.
- 3) **Analysis of eicosanoid production.** We thank Reviewer #2 for comments regarding our methodology to measure the eicosanoid species in the cultured hepatocytes. We agree that

the number of measured eicosanoid species is too small, and some key eicosanoid species were not measured. Thus, we utilized an alternative methodology for eicosanoid measurement, but still could not detect the species mentioned by the reviewer. Consequently, we decided that the data presented in the original Figure 3e is inconclusive, and that it is best to remove it, and all the associated conclusions from the manuscript. Additionally, in response to another comment from Reviewer #2, we performed eicosanoid profiling in the livers of male and female Western Diet-fed mice. The new measurements are presented in **Supplementary Figure 11**. Unlike in cultured hepatocytes, we could detect 19 eicosanoid species with high confidence in Western Diet-fed mouse livers. While the data was variable, we did not detect any differences in the hepatic levels of any measured eicosanoids between *Tlcd1/2* DKO mice and controls. Finally, we discuss the results and limitations of our eicosanoid measurements, and propose this subject to be more comprehensively investigated in future studies in the Discussion section.

We outline the other relevant changes made to the manuscript in the point-by-point rebuttal below.

Response to the specific reviewer points

Reviewer #1 (Remarks to the Author):

Petkevicius and colleagues present a study identifying *TLCD1/2* as novel regulators of cellular PE composition, i.e. PE remodeling. The authors generated *TLCD1/2* KO mice (and cells), and apply various sophisticated omics techniques including lipidomics. Of note, the lipidomics method matches the standards to get reproducible data and most importantly, it is quantitative. The paper is well written, although I think it would benefit from a separation of results and discussion. It would make sense to discuss some specific points in more depth, including the relevance/impact of *TLCD2* for PE composition/remodeling compared to *TLCD1*. I guess, the paper was transferred from another journal, where it was submitted initially as in the letter format. Overall, the methods and analyses applied are sound, the findings are novel and their functional relevance is demonstrated. I support publishing the paper after addressing the following points.

We thank the reviewer for their positive feedback on our work. We have already clarified the points related to the manuscript formatting. We include the point-by-point answers below.

Major points that should be clarified:

1. The basis of the study is the “large scale genome-lipid association map”. The authors should explain at least some methodical basics and details of the analyses for non-experts in genome associations, in the methods section and the Legend of figure 1. Not every reader wants to look deeply into Citation [12], to understand/read the associated Figures 1A and B/method.

As per the reviewer’s suggestion, we have now included a brief description of the methodology in the figure legend, and wrote a brief methods section to reflect the methodology used in the original reference (Linke *et al*).

2. “*Tlcd1* deletion resulting in major reductions in MUFA-containing PE species, while *Tlcd2* deletion

had a similar but lesser effect". Please put this statement into the right perspective. Tlcd2 had almost no effect on PE, except for 1 species (Ext. Figure 1D).

We have now corrected our interpretation in the Results section, and briefly discussed this aspect in the Discussion.

3. I do not understand the sense of Ext. Fig. 4A-B and the statement "To determine whether TLCD1/2 remodel hepatic PEs by regulating gene transcription". Do you mean be regulating e.g. transcription of LPCATs? Please clarify.

We have now clarified this point in the text. As the reviewer indicated, we were referring to potential regulation of LPCAT or the transcription of other lipid remodelling enzyme-encoding genes.

4. I would prefer a simple barplot showing the major PE species (e.g. >1 or 5% of total) instead of the heatmap for Fig.2E. It would more clearly provide the relevant information.

We thank the reviewer for this suggestion. We have now removed the heatmap and instead present the barplots of selected relevant PE species (the ones with the SFA and MUFA at the sn-1 position), expressed as % of total cellular PE. We note that similar to mouse livers, sn-1 MUFA-containing PE species are not usually high in abundance relative to other PE species, and some of them (i.e. PE 16:1_16:1 or PE 16:1_22:6) correspond to approximately 1% of the total cellular PE species.

5. Are there also lipidomic (PE!) data available for the HepG2 lines (over)expressing TLCD1/2? Please include.

We have now performed lipidomics of the PE species in HepG2 lines with stable HA-TLCD1 and HA-TLCD2 expression. New results are included in the **Supplementary Figure 6a**. As expected, the PE species regulated in the TLCD1/2 DKO HepG2 cells exhibit the opposite pattern of regulation to the HepG2 cells with stable expression of HA-TLCD1 or HA-TLCD2. We note that the effect of stable TLCD1 and TLCD2 expression is substantially smaller than the effect of TLCD1/2 deletion. We also discuss that such findings suggest that TLCD1/2 proteins are required, but may not be rate-limiting for the synthesis of MUFA-containing PE species.

6. Please show the expression (e.g. western blots) of some ER and mitochondrial markers, so that it is possible to evaluate the enrichment/contamination of your mitochondrial fraction. Or, show some lipidomic data supporting a high mitochondrial enrichment (in addition to PE =>cardiolipin).

As per the reviewer's suggestion, we have now generated Western blots of the total livers and mitochondria isolated using our protocol. Unfortunately, we utilized all of our original mitochondrial samples for the lipidomics analysis. Therefore, we repeated the mitochondrial isolation on a subset of the mouse livers (4 mice in total, n=2/genotype,) using the exact same protocol, and present the Western blots for ER markers Calnexin and Calreticulin, mitochondrial markers VDAC and COX IV and cytosolic marker GAPDH for the total liver and mitochondrial fraction to demonstrate the relative mitochondrial enrichment and the purity of our isolation. The new results are presented in the new **Supplementary Figure 6d**. We also mention in the text that despite the strong mitochondrial enrichment, some of the ER is still present in the isolated mitochondria.

7. "Phenotyping of TLCD1/2 mice": I would shift this part to the front of the paper, just when you introduce your mouse models.

We thank the reviewer for this suggestion. We have now introduced the general phenotyping of Tlcd1/2 DKO animals as a **Supplementary Figure 3**, right after this mouse model is introduced.

8. Are there already some other factors known influencing the severity of NASH in a sex-specific manner? Please discuss.

We have now included a discussion on sex-specific differences in NASH and some of the known factors that mediate them.

Reviewer #2 (Remarks to the Author):

The manuscript by Petkevicius et al. entitled "TLCD1 and TLCD2 regulate cellular phosphatidylethanolamine composition and promote the progression of non-alcoholic steatohepatitis" presents results supporting a role for Tlcd genes in the pathogenesis of NASH. The authors identified two genes (Tlcd1 and Tlcd2) in chromosome 11 that are associated with the PE composition in liver tissue. Double knockouts (DKO) mice for these genes appeared to show attenuated development of NASH. The study is comprehensive and addresses an important medical subject, but authors need to provide more convincing data on the following two aspects:

We thank the reviewer for acknowledging the importance of our work. We provide the point-by-point responses below.

1. Results shown in Figure 3E about eicosanoid production are not strong enough as rationale for investigation in NASH models, as authors wrote in page 6, lines 7-8. First, eicosanoid production by hepatocytes appears to be in nanograms, which is quite high production. Second, eicosanoids included in the analysis are not biologically active, either are inactive metabolites and not the typical eicosanoids that one could expect to be produced by hepatocytes. I am wondering if authors selected the right platform to measure eicosanoids. Eicosanoids such as PGF2alpha are highly produced by hepatocytes, same for LTB4 metabolites and other lipid mediators derived from EPA and DHA including monohydroxy fatty acids as 18-HEPE and 17-HDHA. Third, changes in eicosanoid production are mild and lack statistical significance. Authors need to perform a more agnostic lipidomic approach including also anti-inflammatory eicosanoids such as LXA4.

We again thank the reviewer for raising this concern. We provide a detailed answer to this point in the General response section of this document.

2. NASH model. Wondering why authors used the western diet model instead of the high fat diet model. Several comments. First, changes are heterogeneous, sometimes differences are observed in males other in females. Second, the model used did not induced insulin resistance. Third, fibrosis needs to be confirmed by Sirius red staining. Fourth, histology needs to be evaluated by a registered pathologist. Other points to consider: ALT levels are unusually low. Also, eicosanoid generating

enzymes (at least COX-1, COX.-2, 5LOX, FLAP,..) and relevant eicosanoids need to be determined in liver tissues.

Taken these two aspects together, authors should remove lines 4-10 in page 7 or provide stronger data supporting the claims.

We provide a detailed response on our choice of NASH model in the General response section of this document, and now include new data from a high-fat diet model in the revised manuscript. We have also removed all the interpretations related to our original eicosanoid analysis from the manuscript (as described in the General response section). We also respond to the specific points raised in this comment as follows:

'changes are heterogeneous, sometimes differences are observed in males other in females' – indeed this is the case, and we typically observe heterogeneity between male and female mice in our *in vivo* studies. Notably, the distribution of the phenotypes observed in this study was not different than our other *in vivo* studies. To address this concern, we have now included a discussion on the potential factor mediating the phenotypic differences between males and females.

'Third, fibrosis needs to be confirmed by Sirius red staining. Fourth, histology needs to be evaluated by a registered pathologist' – we have now performed Picrosirius red (PSR) staining of the Western diet-fed mouse livers and our senior veterinary pathologist scored the images using the Kleiner *et al* method. We present the representative images of PSR staining in **Supplementary Figure 10i**, and the subjective scoring and its non-parametric statistical evaluation in the **Supplementary Figure 10j**.

'ALT levels are unusually low' – we thank the reviewer for spotting this error. We have obtained the original data as $\mu\text{kat/L}$ and did not make a conversion to U/L when making the graphs. We have now corrected this.

'Also, eicosanoid generating enzymes (at least COX-1, COX.-2, 5LOX, FLAP,..) and relevant eicosanoids need to be determined in liver tissues' – the transcripts encoding the eicosanoid generating enzymes are presented in **Figure 4f**. We have also now performed eicosanoid profiling in the livers of WD mice, and many of the products of the reviewer's indicated eicosanoid generating enzymes were measured with high-confidence. We present the new data in **Supplementary Figure 11** and discuss it in the Discussion.

Apart from these two critical issues, other points to consider:

1. DKO mice have decreased MUFA. They also have decreased EPA levels. This needs to be further explored, at least the composition of 18_HEPE and discussed.

We have attempted to measure the levels of 18_HEPE in both cultured hepatocytes and mouse livers but unfortunately were not able to detect it using our methodology. We include this point as a limitation in the Discussion section and propose future work to investigate the role of TLC1/2 proteins in regulating hepatocyte eicosanoid production.

2. Radiolabeled MUFAs accumulate in BAT, a critical fat depot in NASH. BAT needs to be explored in the NASH model.

We thank the reviewer for this suggestion. We have now performed the gene expression analysis in the BAT isolated from *Tlcd1/2* DKO and WT control mice fed both high-fat diet (HFD) and Western diet (WD). We found no differences in the expression of genes involved in fatty acid metabolism and thermogenesis in BAT of either HFD or WD mouse models. To keep the manuscript concise, we chose to only present the data from the HFD study in the new **Supplementary Figure 8h**.

3. HepG2 clones (Fig2E) are unnecessary and do not add much to the final message. Would be more relevant to check human tissues or other human material from patients.

We agree with the reviewer that having data from human tissues would be very informative. In our paper, we present the discovery of TLCD1/2 proteins as regulators of the MUFA-containing PE species, and demonstrate that they function independently of mouse age, diet, sex or feeding state to maintain the hepatic levels of MUFA-containing PE. Furthermore, we do not suggest that they are selectively upregulated in a disease state. However, as we do want to demonstrate that TLCD1/2 function is conserved between mouse and human cells, we believe that the loss of function experiments performed in HepG2 cells are important to address this point. We would propose that the analysis of human tissue and/or material from patients should be performed in future work.

4. What about other phospholipids apart from PE in mitochondria? BMP, PS and cardiolipin.

Thank you for this suggestion. We provide a detailed answer for each of the suggested phospholipid species below:

Cardiolipin: We attempted to measure cardiolipin species in the mitochondrial lipid extracts using a modified lipidomics protocol, to allow detection of higher *m/z* species. Unfortunately, we could only detect 2 CL species with very high variability between the samples. As such, we believe that our lipid extraction protocol may not extract CL species efficiently. However, in the total liver lipidomics dataset generated by an external company Lipotype (Supplementary Data 1c), 10 CL species were measured with high confidence. The levels of the measured CL species were similar between wild-type and *Tlcd1/2* DKO livers. As CL are highly enriched in mitochondria, we believe that the mitochondrial CL composition is unlikely to be affected by *Tlcd1/2* deletion, given there are no changes in the total livers (unlike PE, where similar changes were observed between the total livers and the isolated mitochondria in *Tlcd1/2* DKO mice).

PS: We were able to measure 5 PS species with high confidence in the mitochondrial lipid extracts, and present the data in the new **Supplementary Figure 6e**. The levels of these 5 PS species are similar in the mitochondria isolated from wild-type and *Tlcd1/2* DKO livers.

BMP: While we could detect several mitochondrial BMP species using our lipidomics platform, we could not get a high-confidence MS2 identification of these species – we know that they are BMP lipids, but the exact molecular nature of their acyl chains could not be confirmed. However, none of the BMP species measured with low confidence showed any changes between genotypes (see volcano plot below). As such, we believe that this lipid class is not regulated by TLCD1/2.

Figure. Volcano plot of BMP species measured with low confidence (expressed as molar % of total PL species for each sample) in mitochondria isolated from the livers of wild-type and Tlcd1/2 DKO chow-fed, 3-month-old male mice (n=7 mice/genotype).

Reviewer #3 (Remarks to the Author):

The paper "TLCD1 and TLCD2 regulate cellular phosphatidylethanolamine composition and promote the progression of non-alcoholic steatohepatitis" contributes with a wealth of novel data regarding the biochemical and functional effects of TLCD1 and TLCD2. The functional effects of these two proteins have not previously been well characterized and the paper therefore makes a strong contribution by extensive and valid experiments using e.g. knockouts of the two proteins. Experimental data from models of mice, cultivated human cells and *C. elegans* have been generated using a broad range of appropriate methods. The authors clearly describe the TLCD1/2 knockout models cause reduced levels of monosaturated fatty acid containing PE species, and broadens the characterizations into e.g. PE-influenced inflammation.

We thank the reviewer for their positive feedback on our work.

1. TLCD1/2 is in the manuscript described to have links to mitochondria. Please extend the description and discussion of intracellular localization of TLCD1 and 2 with regard to mitochondria. It seems like TLCD1 and 2 are missing in the extensive (>1000) published compendium of Mitochondrial proteins (Mitocarta, Mootha lab), although they in the present paper co-precipitate with some proteins of inner mitochondrial membranes (ATP5-complex subunits) as well as other

mitochondrial proteins PHB and PHB2. It might be that co-precipitation is mediated by PE of the mitochondrial membrane (?), which would indicate only indirect physical interaction, or by transient interaction during protein import into mitochondria. Please extend discussion of these topics and refer to compendium(s) of mitochondrial proteins.

We agree with the reviewer that this is an important point and have now included an extensive discussion regarding the interaction of TLCD proteins with the mitochondria. As the reviewer indicated, we believe that the interaction is transient, and as seen in our immunofluorescence images, only a subset of mitochondria interact with TLCD proteins at a given time. As such, it might be difficult to identify TLCDs when the mitochondrial proteins are used as bait or when the mitochondrial fraction is isolated by centrifugation (like in the MitoCarta database that the reviewer indicated).

Furthermore, we now include a very recent reference demonstrating that di-monounsaturated PE species in yeast are synthesized in mitochondria (Renne et al, EMBO J, 2022, PMID: 34873731) and discuss its relevance to our findings. Also, we include a reference discussing the role of the closest human TLCD1/2 paralog CLN8, which mediates the interaction between the ER and lysosomes (di Ronza et al, Nature Cell Bio 2018, PMID: 30397314) and discuss the potential similarities between TLCD1/2 and CLN8 as proteins that mediate the molecular transport between the two organelles.

2. Figure 3B describes Ingenuity Pathway analysis (IPA) data, in a rather superficial way. Assure that more detailed data of each functional sub-group are accessible to the reader, i.e. number of proteins and their identities, and possibly the size of background dataset (genomewide or practical cellular proteome?).

We have now included a detailed breakdown of the pathway analysis presented in Figure 3B – the list of corresponding protein hits associated with each pathway, their fold enrichment in pulldowns and p values. We include the new data as a table in **Supplementary Data 2c**. We also specify in the Figure legend that the pathway analysis was conducted on the genome-wide dataset.

Minor:

Page 5 line 2. A description of the abbreviation “HA” is lacking in the manuscript and could be added in the context of sentence “cell lines expressing HA-tagged TLCD1 or 3 TLCD2 proteins....”. Likewise could also the abbreviation “GFP” be described.

We thank the reviewer for raising this point. We have now included the full nomenclature for HA and GFP in the text, when they are first mentioned.

Reviewers' Comments:

Reviewer #1:

Remarks to the Author:

The authors provided all the information requested and improved the manuscript. I have no further demands and clearly support publishing the study.

Reviewer #2:

Remarks to the Author:

The authors significantly improved the paper by removing the eicosanoid part, which was not fully supported by the data. However, this reviewer is still puzzled that they did not detect most lipid mediators that are produced by liver cells and or liver tissue, such as 18-HEPE. The authors reply very superficially to this comment and just mentioned in the general response to responses that "we utilized an alternative methodology for eicosanoid measurement, but still could not detect the species mentioned by the reviewer". Authors need to be more precise and properly address this comment. Which methodology is alternative? What is it? Need to define.

If the authors are not able to fully detect eicosanoid production nor the monohydroxy fatty acid intermediates, they need to at least provide PUFA levels including AA, EPA and DHA.

Reviewer #3:

Remarks to the Author:

This reviewer still think it is important to better specify the description in the manuscript about the physical association of TLCs to mitochondria. The data of the manuscript describe interesting links to mitochondria, however it is important to clarify that TLCs aren't known to be intra-mitochondrial proteins, since mitochondrial compendia don't contain TLC1 or 2. And these compendia are based on numerous independent datasets.

“TLCD1 and TLCD2 regulate cellular phosphatidylethanolamine composition and promote the progression of non-alcoholic steatohepatitis” – Second response to reviewers

REVIEWER COMMENTS

Reviewer #1 (Remarks to the Author):

The authors provided all the information requested and improved the manuscript. I have no further demands and clearly support publishing the study.

Reviewer #2 (Remarks to the Author):

The authors significantly improved the paper by removing the eicosanoid part, which was not fully supported by the data. However, this reviewer is still puzzled that they did not detect most lipid mediators that are produced by liver cells and or liver tissue, such as 18-HEPE. The authors reply very superficially to this comment and just mentioned in the general response to responses that "we utilized an alternative methodology for eicosanoid measurement, but still could not detect the species mentioned by the reviewer". Authors need to be more precise and properly address this comment. Which methodology is alternative? What is it? Need to define.

We apologize for the lack of clarity in our initial response. We used UPLC-ESI-MS/MS (ultra-performance liquid chromatography-electrospray ionization-tandem mass spectrometry) method for all eicosanoid measurements presented in both the initial and the revised versions of our manuscript. These measurements were performed as a service by Swedish Metabolomics Centre, which has an established method in measuring oxylipin species in cells and tissues (PMID: 26115647). We note that their method is not specifically tailored to measure eicosanoid species produced by liver cells, as such, some of the hepatocyte-derived eicosanoids are missing from the analysis.

In the attempt to address the initial concerns from the Reviewer #2 by measuring more eicosanoid species, we utilised our internal LC-MS/MS (liquid chromatography-tandem mass spectrometry) platform, which we routinely use for standard lipidomics analyses (see Methods section for details), that we referred to as ‘an alternative methodology’. However, with the limited resources allocated to this project, we could not establish an LC-MS/MS protocol for a high-confidence measurement of the eicosanoid species mentioned by the Reviewer #2 in their initial comment.

If the authors are not able to fully detect eicosanoid production nor the monohydroxy fatty acid intermediates, they need to at least provide PUFA levels including AA, EPA and DHA.

We thank the reviewer for this suggestion. We have now determined the levels of AA, EPA and DHA containing PC and PE species in the livers of WD-fed male and female mice. We present the new data in the **Supplementary Fig. 11b**, and briefly discuss our findings in the Discussion.

Reviewer #3 (Remarks to the Author):

This reviewer still think it is important to better specify the description in the manuscript about the physical association of TLCDs to mitochondria. The data of the manuscript describe interesting links to mitochondria, however it is important to clarify that TLCDs aren't known to be intra-mitochondrial proteins, since mitochondrial compendia don't contain TLCD1 or 2. And these compendia are based on numerous independent datasets.

We thank the reviewer for further reiterating this point. As per reviewer's suggestion, we have now clearly indicated that TLCD1 and TLCD2 are not intra-mitochondrial proteins in the Results section, where the physical association of TLCDs to mitochondria is described. We reference three different manually curated mitochondrial proteome databases to support this statement. We have also modified the Discussion on this point accordingly.

Reviewers' Comments:

Reviewer #2:

Remarks to the Author:

Authors have satisfactorily addressed my comments.

“TLCD1 and TLCD2 regulate cellular phosphatidylethanolamine composition and promote the progression of non-alcoholic steatohepatitis” – Final response to reviewers

REVIEWERS' COMMENTS

Reviewer #2 (Remarks to the Author):

Authors have satisfactorily addressed my comments.

We thank the reviewer for all their help in improving our manuscript.